# Towards the Dynamics of Representation Changes during Parameter Pruning for Network Compression

## Abstract

This study explores the dynamics of interactions encoded by deep neural networks (DNNs) when we prune network parameters with an increasing pruning ratio. We discover a three-phase dynamics of the generalizability of the interactions removed by the parameter pruning operation, which clarifies a central issue in symbolic generalization, i.e., how interactions serve as the underlying factors that determine the change of a DNN's performance. Experimental results demonstrate that the pruning operation mainly removes high-order interactions at low pruning ratios. Because the removed high-order interactions are usually unlikely to generalize, the removal of high-order interactions has a negligible impact on testing performance. In contrast, under higher pruning ratios, both low-order and high-order interactions are gradually removed. The high generalizability of the removed low-order interactions leads to a noticeable decline in testing performance.

## 1 Introduction

In the field of post-hoc explanation of deep neural networks (DNNs), there has been a trend toward more precise and nuanced approaches (Li & Zhang, 2023b; Ren et al., 2024a). Mechanistic interpretability studies (surveyed in Appendix A) are developed to explain individual neurons. In comparison, another emerging direction, namely *symbolic generalizability*, has provided a new strategy to explain the generalizability of DNNs (Kang et al., 2024; Ren et al., 2023a; 2024a; Tsai et al., 2023). As Figure 1 shows, this theory aims to define and use the generalizability of inference patterns encoded in a DNN to explain the generalizability of the entire DNN.

**Background: symbolic generalizability.** Recent advances in symbolic generalizability (surveyed in Appendix A) have revealed a counterintuitive phenomenon: **the complex inference logic of a DNN can be precisely explained by a small set of *AND-OR interactions* in mathematics.** Given an input sample, each interaction represents an AND relationship (or an OR relationship) among input variables that is equivalently encoded by the DNN. For instance as illustrated in Figure 1, the large language model encodes an AND interaction among input words $S$={"Electric", "currents", "around"}. If and only if all words in $S$ appear in the input prompt, this interaction is triggered and contributes an effect of 0.45 to boost the confidence of generating the target word "conductors." Masking any words in $S$ will deactivate the interaction and remove its effect. It is proven by Chen et al. (2024) that **people can use numerical effects of such interactions to accurately predict the DNN's classification scores on exponentially many diverse samples**, which guarantees the faithfulness of such explanation.

Therefore, the next fundament issue in this field is to use the generalizability of the **compositional interactions** to explain the generalizability of the **entire DNN**[1] (Zhang et al., 2024; Zhou et al., 2024; Ren et al., 2024b; Cheng et al., 2025). For example, as Figure 2 shows, the interaction among image patches of "wing" consistently appears in both training samples and testing samples, and contributes similar interaction effects. Therefore, this interaction is considered generalizable. It is found that interactions in a highly generalizable DNN are more likely to generalize to testing samples than those in a less well-trained DNN (Zhou et al., 2024).

---

[1]It is because Eq. (3) shows the DNN's output $v(\boldsymbol{x})$ can be represented as the sum of interaction effects.

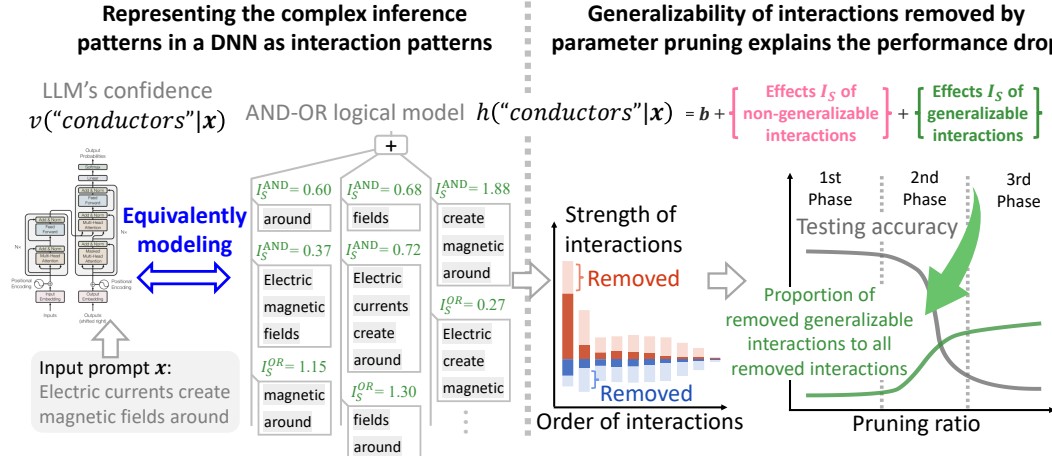

Figure 1: (left) It is proven (Ren et al., 2024a) that the intricate inference logic can be mathematically explained by a small set of AND-OR interactions. (right) Some interactions are removed by parameter pruning, and the distribution of the removed interactions exhibits a specific three-phase dynamics w.r.t. its complexity and generalizability, when the pruning ratio progressively increases. **Please see Appendix I for examples of AND-OR logical models that explains LLMs.**

**Our work.** However, previous findings are mostly based on empirical observations, e.g., Zhou et al. (2024) found that low-order interactions often exhibited stronger generalizability than high-order interactions. In contrast, we aim to **extend previous descriptive studies to interventional studies, in order to use the change of interactions' generalizability to explain the underlying mechanisms for the change of a DNN's performance**.

Specifically, in this study, we focus on parameter pruning (Han et al., 2016; Fang et al., 2023), a classical strategy for removing trivial parameters from neural networks and compressing feature representations. Therefore, this paper investigates how parameter pruning operations affect the representation quality of DNNs when the pruning ratio progressively increases.

This paper discovers a specific three-phase dynamics of DNNs' interaction changes under progressively increaseing pruning ratio. **(1)** In the first phase, the compression ratio is low, and the parameter pruning operation only removes high-order interactions while preserving low-order ones. However, the testing accuracy is still not significantly affected, which can be explained by the finding in (Zhou et al., 2024; Cheng et al., 2025) that most high-order interactions represent noise patterns that cannot generalize to testing samples. **(2)** In the second phase, the further increase of the pruning ratio starts to remove low-order interactions. The high generalizability of the removed low-order interactions leads to a noticeable drop in testing accuracy. **(3)** In the third phase, most interactions have been removed, and even higher pruning ratios would not substantially affect the performance.

Our work successfully demonstrates the strong potential of symbolic generalization by showing that the generalizability of interactions can effectively explain the root causes behind the performance degradation of a DNN, in the scenario of neural network compression. The removal of low-order interactions tends to cause significant degradation of DNNs' testing performance. On the contrary, the removal of high-order interactions has little impact the DNNs' testing performance. In addition, our discovery also provides a more rigorous validation for prior empirical findings in the field of symbolic generalization (Zhou et al., 2024).

**Practical value.** Based on our findings, in follow-up studies, we penalize the strength of all non-generalizable interactions during the training of the DNN. This method allows the DNN to maintain its testing performance at higher pruning ratios. See Appendix B for details.

In summary, the discovery of the above three-phase dynamics provides a new perspective to understand the underlying factors that hurt the DNN's performance when the DNN's parameters are pruned. Compared to traditional enginnering settings for parameter pruning, the rigorous mechanistic explanation provides a clear metric to determine the performance limits of parameter pruning.

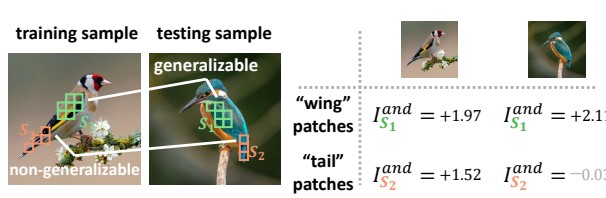

Figure 2: The interaction among "wing" patches consistently contributes similar interaction effects, thus being considered generalizable. The interaction among "tail" patches cannot generalize to testing samples, thus being considered non-generalizable.

# 2 DYNAMICS OF REPRESENTATION CHANGES DURING PARAMETER PRUNING

## 2.1 PRELIMINARIES: INTERACTIONS

Given an input sample with $n$ input variables (indexed by the set $N = \{1, 2, \ldots, n\}$), denoted by $\boldsymbol{x} = [x_1, x_2, \cdots, x_n]^T$, we use $v(\cdot)$ to represent the scalar output of the DNN. There exists different settings for the scalar output. Typically, for multi-category classificaiton, we follow Deng et al. (2022) to set the widely used classification confidence as follows.

$$v(\boldsymbol{x}) \stackrel{\text{def}}{=} \log \frac{p(y = y^* \mid \boldsymbol{x})}{1 - p(y = y^* \mid \boldsymbol{x})} \tag{1}$$

where $p(y = y^* \mid \boldsymbol{x})$ represent the predicted classification probability on the ground-truth category.

**Problem setting.** As an emerging explanation direction (surveyed in Appendix A), symbolic generalizability studies (Li & Zhang, 2023b; Ren et al., 2024a; Chen et al., 2024; Cheng et al., 2025) are proposed to explain the generalizability of primitive inference patterns encoded by a DNN. The basic idea is to use a logical model $g(\cdot)$ to explain inference patterns in the DNN $v(\cdot)$, and the faithfulness of explanation is ensured by requiring a sufficently concise logical model $g$ to accurately explain a sufficiently large number of network outputs $v$ w.r.t. the following two requirements. **(1) Fidelity requirement:** the logical model $g(\cdot)$ is powerful enough to well match the network outputs $v(\cdot)$ on all diverse samples in a sufficient large set $\boldsymbol{\Psi}$. **(2) Conciseness requirement:** the logical model is supposed to only consist of a small number of inference patterns. These requirements can be formally expressed as

$$\forall \boldsymbol{x}' \in \boldsymbol{\Psi}, \ g(\boldsymbol{x}') = v(\boldsymbol{x}') \quad \text{subject to} \quad \text{complexity}(g) \leq M, \tag{2}$$

where $M$ is the upper complexity bound of the logical model $g(\cdot)$.

**First, the logical model $g(\cdot)$ is implemented to encode AND-OR interactions among the input variables in $N$ (Li & Zhang, 2023b; Chen et al., 2024).** Figure 1 depicts how each interactions corresponds to a non-linear relationship among a set of input variables. The formal definition of the model is given below. **See Appendix I for examples of logical models that explains LLMs. A video demo that introduces the logical model is also attached as a supplementary material.**

$$g(\boldsymbol{x}') \stackrel{\text{def}}{=} \sum_{S \in \Omega^{\text{and}}} I_S^{\text{and}} \cdot \delta_{\text{and}}(S \mid \boldsymbol{x}') + \sum_{S \in \Omega^{\text{or}}} I_S^{\text{or}} \cdot \delta_{\text{or}}(S \mid \boldsymbol{x}') + b \tag{3}$$

*The AND trigger function $\delta_{and}(S \mid \boldsymbol{x}')$ activates (returns 1) if and only if all variables in the subset $S \subseteq N$ are present in $\boldsymbol{x}'$; if any variable in $S$ is masked[2], the function returns 0. $I_S^{\text{and}}$ is the scalar weight of the AND interaction $S$. Similarly, *the OR trigger function $\delta_{or}(S \mid \boldsymbol{x}')$ activates whenever at least one variable in the subset $S$ is present. $I_S^{\text{or}}$ is the scalar weight of the OR interaction $S$. $\Omega^{\text{and}}$ and $\Omega^{\text{or}}$ denote the set of AND interactions and that of OR interactions, respectively, extracted from the input $\boldsymbol{x}'$. $b$ is a scalar bias term.

**Second, the fidelity requirement is guaranteed by the universal matching property in Theorem 2.1.** Under a specific settings of weights, the logical model $g(\cdot)$ can accurately reproduce the outputs of the DNN $v(\cdot)$, no matter how we augment the input $\boldsymbol{x}$ by randomly masking a subset of input variables. In other words, the sample set $\boldsymbol{\Psi} = \{\boldsymbol{x}_T \mid T \subseteq N\}$ contains $2^n$ masked states, which is sufficiently large. Here, $\boldsymbol{x}_T$ denotes the masked version of the sample $\boldsymbol{x}$ that retains only the variables in $T$, with others in $N \setminus T$ masked[2].

---

[2]The masking of the $i$-th input variables is implemented by setting $x_i$ to the baseline values $b$, which is commonly set as the average value of the variable on multiple samples (Dabkowski & Gal, 2017).

**Theorem 2.1** (*Universal matching property, proven by (Chen et al., 2024) and Appendix G) Given a sample $\boldsymbol{x}$ and a DNN $v(\cdot)$, let us set the scalar weights $I_S^{and}$ and $I_S^{or}$ in $g(\cdot)$ as $\forall S \subseteq N$, $I_S^{and} = \sum_{T \subseteq S} (-1)^{|S|-|T|} \cdot u_T^{and}$, $I_S^{or} = -\sum_{T \subseteq S} (-1)^{|S|-|T|} \cdot u_{N \backslash T}^{or}$, subject to $u_T^{and} = 0.5 \cdot v(\boldsymbol{x}_T) + \gamma_T$ and $u_T^{or} = 0.5 \cdot v(\boldsymbol{x}_T) - \gamma_T$. bias $b = v(\boldsymbol{x}_\emptyset)$, and $\{\gamma_T\}$ is a set of learnable parameters. Then we have $\forall T \subseteq N$, $v(\boldsymbol{x}_T) = g(\boldsymbol{x}_T)$.*

**Interaction extraction.** We adopt the method of Chen et al. (2024) to learn parameters $\{\gamma_T\}$, so as to extract a set of AND-OR interactions. Detailed pseudocode are provided in Appendix F.

**Third, the conciseness requirement is ensured by the sparsity property of interactions.** Ren et al. (2024a) have shown that a well-trained DNN encodes only a relatively small number of salient interactions, roughly $O(n^p/\tau)$ with $p \in [1.5, 2]$, under three common conditions introduced in Appendix E. We can construct a much more compact model $\hat{g}(\cdot)$ with the small number of salient interactions in $\Omega^{and} = \{S \subseteq N : |I_S^{and}| > \tau\}$ and $\Omega^{or} = \{S \subseteq N : |I_S^{or}| > \tau\}$. Thus, we can accurately predict the network outputs over differently masked inputs. $\tau$ is a scalar threshold, which is set to $\tau = 0.05 \cdot \sum_{\boldsymbol{x}} |v(\boldsymbol{x}_T) - v(\boldsymbol{x}_\emptyset)|$.

**Order of interactions.** The order of an AND/OR interaction $S$, written as $\text{order}(S) = |S|$, is defined by the quantity of variables in $S$. It serves as an indicator of the interaction's complexity.

## 2.2 Explaining the generalizability of DNNs in terms of interactions

As introduced in Section 2.1, interactions provide a new perspective for explaining the root causes for the performance of DNNs. Prior studies (Zhou et al., 2024) have found that the generalizability of the **entire** DNN can be explained by the overall generalizability of its **compositional** interactions[1]. In particular, it has been found (Zhou et al., 2024; Liu et al., 2023) that the *complexity* and the *generalizability* of interactions are the key factors that determine network performance.
•Zhou et al. (2024) found that **interactions of higher orders were usually less generalizable.**
•Liu et al. (2023) found that **high-order interactions showed more vulnerability to feature noises.**
•**The generalizability of interactions could explain the overfitting of DNNs** (Cheng et al., 2025).

Despite above achievements, existing studies (Ren et al., 2023a; Li & Zhang, 2023b; Ren et al., 2024a; Chen et al., 2024; Cheng et al., 2025) mainly passively and empirically explained the generalizability and complexity of interactions, and they still failed to *investigate the precise relationship between network performance and interaction patterns in terms of interventional studies.*

**Therefore, in this study, we focus on neural network parameter pruning, and aim to explore how the complexity and generalizability of interactions encoded by a DNN are affected when we apply a progressively increasing pruning ratio to it.** Furthermore, we aim to uncover the underlying mechanisms behind DNNs' performance changes under parameter pruning. It is because parameter pruning is a classical strategy for removing trivial parameters from neural networks, which provides a valuable perspective for analyzing neural network behaviors.

**Quantifying the complexity of interactions.** Definition 2.1 uses the distribution of interaction strength over different orders to measure a DNN's representational complexity (see Figure 3).

**Definition 2.1** *Given a set of AND interactions $\hat{\Omega}^{and}$ and a set of OR interactions $\hat{\Omega}^{or}$, the total strength of all $m$-order positive interactions $\mathbf{I}^{(m),+}(\hat{\Omega}^{and}, \hat{\Omega}^{or})$ and the total strength of all $m$-order negative interactions $\mathbf{I}^{(m),-}(\hat{\Omega}^{and}, \hat{\Omega}^{or})$ are defined as follows:*

$$\mathbf{I}^{(m),+}(\hat{\Omega}^{and}, \hat{\Omega}^{or}) = \sum_{S \in \hat{\Omega}^{and}:|S|=m} \max(I_S^{and}, 0) + \sum_{S \in \hat{\Omega}^{or}:|S|=m} \max(I_S^{or}, 0),$$
$$\mathbf{I}^{(m),-}(\hat{\Omega}^{and}, \hat{\Omega}^{or}) = \sum_{S \in \hat{\Omega}^{and}:|S|=m} \min(I_S^{and}, 0) + \sum_{S \in \hat{\Omega}^{or}:|S|=m} \min(I_S^{or}, 0). \tag{4}$$

**Quantifying the generalizability of interactions.** According to the original definition (Zhou et al., 2024), the interaction $S$ is considered generalizable if it consistently contributes similar salient interaction effects across different testing samples. However, it requires to check interactions through all testing samples, which is computational infeasible. Thus, we adopt an approximate yet more efficient method (He et al., 2025) to identify generalizable interactions.

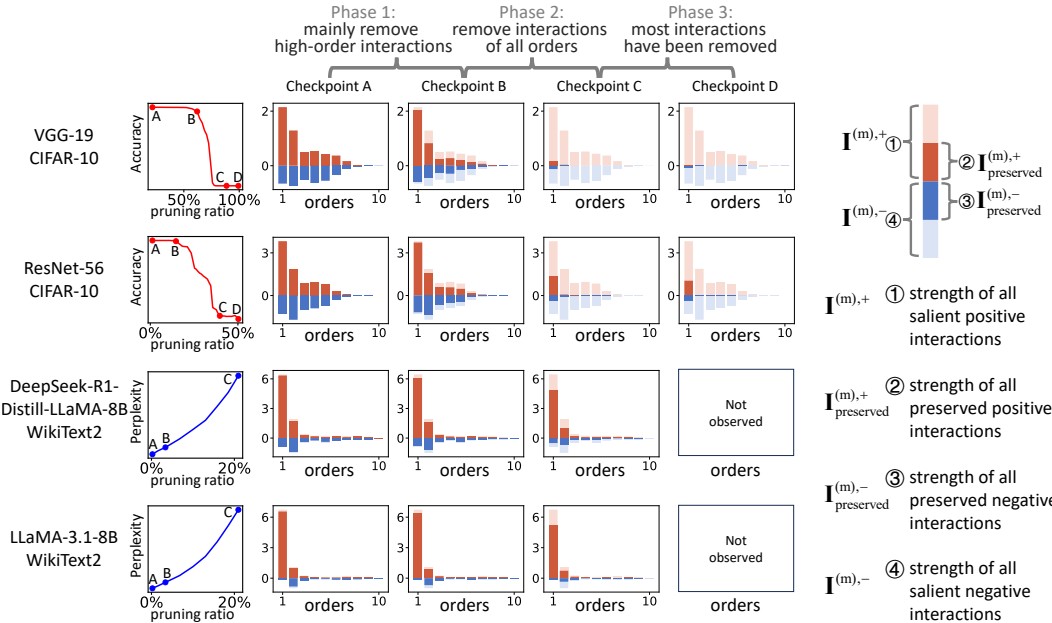

Figure 3: Changes of the distributions of $\mathbf{I}^{(m),+}$, $\mathbf{I}^{(m),-}$, $\mathbf{I}_{\text{preserved}}^{(m),+}$, and $\mathbf{I}_{\text{preserved}}^{(m),-}$ when the pruning ratio increases. These changes can be divided into three phases. **See Appendix J for more results.**

Specifically, we train a separate DNN on testing samples, referred to as a *reference DNN*. An interaction S is defined generalizable if it is also utilized by this reference DNN for inference. This is because all interactions encoded by the reference DNN are learned from the testing samples.

**Definition 2.2** *Let us be given a salient AND interaction S extracted by a DNN $v(\cdot)$ from the input sample $\boldsymbol{x}$, subject to $|I_S^{and}| > \tau$. If this AND interaction is also extracted as a salient interaction by reference DNN $v^*(\cdot)$, and produces a consistent effect (i.e., $|I_S^{*,and}| > \tau$ and $I_S^{and} \cdot I_S^{*,and} > 0$), then we consider this interaction generalizable. The generalizability of an OR interaction is defined similarly. Generalizable AND/OR interactions can be identified by the following two binary metrics:*

$$G_S^{and} = \mathbb{1}(|I_S^{*,and}| > \tau \ \ and \ \ I_S^{*,and} \cdot I_S^{and} > 0), \ \ G_S^{or} = \mathbb{1}(|I_S^{*,or}| > \tau \ \ and \ \ I_S^{*,or} \cdot I_S^{or} > 0) \quad (5)$$

*where $G_S^{and} \in \{0, 1\}$ and $G_S^{or} \in \{0, 1\}$. $\mathbb{1}(\cdot)$ is an indicator function. It returns 1 if the condition is true, and returns 0 otherwise. $I_S^{*,and}$ and $I_S^{*,or}$ represent the numerical effects of the AND interaction S and the OR interaction S, respectively, extracted by the reference DNN.*

### 2.3 THREE-PHASE DYNAMICS OF INTERACTIONS DURING PROGRESSIVE PRUNING

In this paper, we explore the dynamics of interactions. When we progressively increase the pruning ratio applied to a DNN, we discover a distinct three-phase dynamics of interaction complexities (orders), which well explains the performance change of the pruned DNN.

**Quantifying the removed and preserved interactions after parameter pruning.** To this end, we propose a set of metrics to quantify the change of interactions after parameter pruning. Given an original DNN $v(\cdot)$ and a pruned DNN $v'(\cdot)$, we follow (Chen et al., 2024) to extract two sets of interaction weights $\{I_S^{and}, I_S^{or} \mid S \subseteq N\}$ and $\{I_S'^{and}, I_S'^{or} \mid S \subseteq N\}$ from each model, respectively. Then, we define the interactions removed after parameter pruning as follows.

**Definition 2.3** *Let us be given a salient AND interaction S extracted by the original DNN $v(\cdot)$ from the input $\boldsymbol{x}$, subject to $|I_S^{and}| > \tau$. If this interaction is no longer salient in the pruned DNN $v'(\cdot)$ (subject to $|I_S'^{and}| \leq \tau$) or exhibits a contrary effect ($I_S'^{and} \cdot I_S^{and} < 0$), then this interaction is*

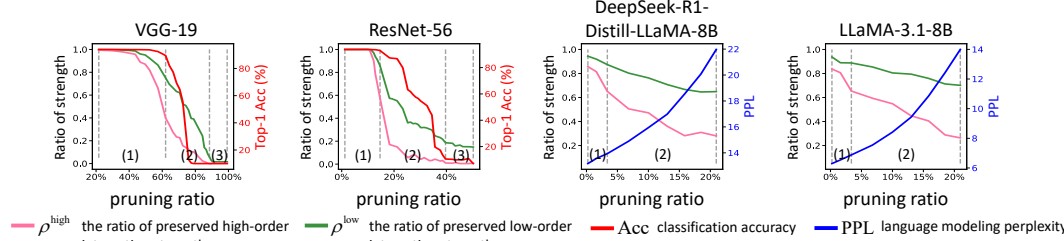

Figure 4: Changes of the ratio of preserved low-order interaction strength $\rho^{\text{low}}$, the ratio of preserved high-order interaction strength $\rho^{\text{high}}$, classification accuracy (or language modeling perplexity) when the pruning ratio increases. **Please see Appendix J for results on more DNNs.**

*considered to have been removed by the parameter pruning operation. Thus, we use the following binary metrics to identify the removed AND/OR interactions.*

$$R_S^{and} = \mathbb{1}(|I'^{and}_S| \leq \tau \;\; or \;\; I'^{and}_S \cdot I^{and}_S < 0), \quad R_S^{or} = \mathbb{1}(|I'^{or}_S| \leq \tau \;\; or \;\; I'^{or}_S \cdot I^{or}_S < 0) \tag{6}$$

Thus, let us use $\Omega^{\text{and}} = \{S \subseteq N : |I_S^{\text{and}}| > \tau\}$ and $\Omega^{\text{or}} = \{S \subseteq N : |I_S^{\text{or}}| > \tau\}$ to denote the sets of salient AND-OR interactions extracted from the orginal DNN $v(\cdot)$. Accordingly, we can use $\Omega_{\text{preserved}}^{\text{and}} = \{S \in \Omega^{\text{and}} : R_S^{\text{and}} = 0\}$ and $\Omega_{\text{preserved}}^{\text{or}} = \{S \in \Omega^{\text{or}} : R_S^{\text{or}} = 0\}$ to denote the preserved AND-OR interactions extracted from the pruned DNN $v'(\cdot)$.

**Evaluation metric.** We measure the ratio of low-order interactions (1-st to 3-rd order) that are preserved after parameter pruning, denoted by $\rho^{\text{low}}$, and the ratio of the preserved high-order interactions (4-th to $n$-th order), denoted by $\rho^{\text{high}}$, to evaluate the representation quality of the pruned DNN. This is because, according to Section 2.2 and Liu et al. (2023); Zhou et al. (2024), high-order interactions are less generalizable to testing samples and more sensitive to feature noises than low-order interactions. Thus, if the pruning operation mainly removes high-order interactions (causing a low $\rho^{\text{high}}$ value), then such removal usually has little effect on the DNN's performance, because most removed interactions are offsetting high-order interactions. Conversely, if the pruning operation mainly removes low-order interactions (causing a low $\rho^{\text{low}}$ value), the DNN's performance will be significantly degraded.

$$\rho^{\text{low}} = \frac{\sum_{\text{type} \in \{\text{and,or}\}} \sum_{S \in \Omega^{\text{type}} : 1 \leq |S| \leq 3} |I_S^{\text{type}}| \cdot (1 - R_S^{\text{type}})}{\sum_{\text{type} \in \{\text{and,or}\}} \sum_{S \in \Omega^{\text{type}} : 1 \leq |S| \leq 3} |I_S^{\text{type}}|}$$
$$\rho^{\text{high}} = \frac{\sum_{\text{type} \in \{\text{and,or}\}} \sum_{S \in \Omega^{\text{type}} : 4 \leq |S| \leq n} |I_S^{\text{type}}| \cdot (1 - R_S^{\text{type}})}{\sum_{\text{type} \in \{\text{and,or}\}} \sum_{S \in \Omega^{\text{type}} : 4 \leq |S| \leq n} |I_S^{\text{type}}|} \tag{7}$$

Figure 3 compares the distribution of interaction strength over different orders between those extracted from the original DNN $v(\cdot)$ (measured by $\mathbf{I}^{(m),+} = \mathbf{I}^{(m),+}(\Omega^{\text{and}}, \Omega^{\text{or}})$ and $\mathbf{I}^{(m),-} = \mathbf{I}^{(m),-}(\Omega^{\text{and}}, \Omega^{\text{or}})$ ) and those extracted from the pruned DNN $v'(\cdot)$ (measured by $\mathbf{I}_{\text{preserved}}^{(m),+} = \mathbf{I}^{(m),+}(\Omega_{\text{preserved}}^{\text{and}}, \Omega_{\text{preserved}}^{\text{or}})$ and $\mathbf{I}_{\text{preserved}}^{(m),-} = \mathbf{I}^{(m),-}(\Omega_{\text{preserved}}^{\text{and}}, \Omega_{\text{preserved}}^{\text{or}})$ ). Figure 4 shows the changes of the ratio of low-order interaction strength that are preserved after pruning $\rho^{\text{low}}$ and the ratio of preserved high-order interaction strength $\rho^{\text{high}}$ when the pruning ratio increases, along with the testing performance of DNNs (measured by classification accuracy or language modeling perplexity). Based on the above experimental results, we can divide the entire dynamics of interactions into three phases.

•*Phase 1:* As Figure 3 and Figure 4 shows, when the pruning ratio is low, only high-order interactions are removed, while low-order ones remain largely unaffected (*i.e.*, $\rho^{\text{high}}$ noticeably decreases but $\rho^{\text{low}} \approx 1$). Note that the removal of high-order interactions usually does not affect the testing accuracy, which has been supported by multiple lines of evidence. (1) Zhou et al. (2024) have found that high-order interactions are less generalizable to testing samples than low-order interactions. (2) The interactions removed in this phase exhibit a spindle-shaped distribution over different orders. A half of these removed interactions produce positive effects and boost the classification score, while the other half produce negative effects and reduce the classification score.

Table 1: Analysis of three-phase dynamics of interactions.

| Perspectives | Phase 1 | Phase 2 | Phase 3 |
|---|---|---|---|
| Testing accuracy | barely affected | significantly decreased | remains low |
| Low-order interactions | slightly removed | significantly removed | most have been removed |
| High-order interactions | significantly removed | significantly removed | most have been removed |
| Genralizability of the removed | slightly increase | significantly increase | most have been removed |
| Genralizability of the emerged | remains low | remains low | remains low |

•*Phase 2:* When the pruning ratio further increases, low-order interactions are also removed besides the removal of high-order interactions. The testing accuracy begins to decrease. The decrease of the testing accuracy is caused by the removal of low-order interactions, because most low-order interactions have been found by Zhou et al. (2024) to represent well-trained inference patterns that can generalize to testing samples.

•*Phase 3:* When the pruning ratio increases to even higher levels, both low-order and high-order interactions show only minor changes because most of them have already been removed. The testing accuracy remains extremely low under excessive parameter pruning.

In sum, when the pruning ratio is low (during Phase 1), parameter pruning mainly removes high-order interactions, which barely affects the performance of DNNs. In contrast, when the pruning ratio is high (during Phases 2 and 3), low-order interactions start to be removed, which explains the performance degradation of DNNs. This discovery provides further support for the assumption in prior work on symbolic generalizability (Zhou et al., 2024; Cheng et al., 2025).

**Settings for network pruners.** We conducted experiments on eight DNNs with three network prunners. The eight DNNs included two LLMs for language generation, two ViT models, and four CNNs for image classification. Considering the adaptability of pruning algorithms to models, (1) we applied LLM-Pruner (Ma et al., 2023) to two different LLMs, including DeepSeek-R1-Distill-LLaMA-8B model (DeepSeek-AI et al., 2025) and LLaMA-3.1-8B model (Grattafiori et al., 2024); (2) we applied Isomorphic-Pruning (Fang et al., 2024) to two residual networks and two ViT models pretrained on the ImageNet-1k dataset (Russakovsky et al., 2015); (3) we applied DepGraph (Fang et al., 2023) to two small networks for image classification on the CIFAR-10 dataset (Krizhevsky, 2009). **Please refer to Appendix H for detailed settings and Appendix J for detailed results.**

## 2.4 UNDERSTANDING THE THREE-PHASE DYNAMICS IN TERMS OF GENERALIZABILITY

In this subsection, we aim to explore the three-phase dynamics of the generalizability of interactions. The precise and fine-grained explanation of detailed performance changes of a DNN presents the core challenge for symbolic generalization.

Specifically, the dynamics of generalizability of interactions can well explain the change of generalizability of the DNN. Table 1 shows the analytical results of the three-phase dynamics of interactions from multiple perspectives. Detailed results can be found in Appendix J.

**Distribution of generalizable interactions.** Based on Definition 2.1, the distribution of generalizable interactions across each $m$-th order can be quantified by the total strength of all positive generalizable interactions $\mathbf{I}_{\text{generalizable}}^{(m),+} = \mathbf{I}^{(m),+}(\Omega_{\text{generalizable}}^{\text{and}}, \Omega_{\text{generalizable}}^{\text{or}})$, and the total strength of all negative generalizable interactions $\mathbf{I}_{\text{generalizable}}^{(m),-} = \mathbf{I}^{(m),-}(\Omega_{\text{generalizable}}^{\text{and}}, \Omega_{\text{generalizable}}^{\text{or}})$, where $\Omega_{\text{generalizable}}^{\text{and}} = \{S \in \Omega^{\text{and}} : G_S^{\text{and}} = 1\}$ and $\Omega_{\text{generalizable}}^{\text{or}} = \{S \in \Omega^{\text{or}} : G_S^{\text{or}} = 1\}$ denote the sets of generalizable AND interactions and generalizable OR interactions, respectively.

Figure 26 in Appendix K shows the distributions of $\mathbf{I}^{(m),+}$, $\mathbf{I}^{(m),-}$, $\mathbf{I}_{\text{generalizable}}^{(m),+}$ and $\mathbf{I}_{\text{generalizable}}^{(m),-}$ over different orders, which are extracted from eight DNNs. Most generalizable interactions are low-order interactions, and most high-order interactions cannot generalize to the reference DNN. Thus, these high-order interactions are considered as non-generalizable patterns.

**Exploring the efficiency of parameter pruning in terms of generalizability of the removed interactions.** We investigate generalizability of interactions removed by parameter pruning, and

Table 2: Efficiency of parameter pruning in the three-phases. See Appendix J for more results.

| Architecture | Phase 1 | Phase 2 | Phase 3 |
|---|---|---|---|
| ResNet-56 | $e^{1.21 \to 14.65} = 0.92$ | $e^{14.65 \to 39.68} = 0.78$ | $e^{39.68 \to 50.27} = 0.38$ |
| VGG-19 | $e^{21.64 \to 62.02} = 0.89$ | $e^{62.02 \to 88.54} = 0.61$ | $e^{88.54 \to 99.23} = 0.36$ |
| DeepSeek-R1-Distill-LLaMA-8B | $e^{0.23 \to 3.37} = 0.78$ | $e^{3.37 \to 20.96} = 0.75$ | — |
| LLaMA-3.1-8B | $e^{0.23 \to 3.37} = 0.74$ | $e^{3.37 \to 20.96} = 0.70$ | — |

thereby analyze the detailed utility of a neural network compression method. In the first phase, pruning mainly removes non-generalizable interactions. In the second and third phases, it mainly removes generalizable interactions, which affects the DNNs' performance. To verify this, let us first quantify the total strength of all removed interactions $\mathbf{A}_{\text{removed}}$, as well as the strength of removed generalizable interactions $\mathbf{G}_{\text{removed}}$.

$$\mathbf{A}_{\text{removed}} = \sum_{\substack{\text{type} \in \\ \{\text{and,or}\}}} \sum_{S \in \Omega^{\text{type}}} |I_S^{\text{type}}| \cdot R_S^{\text{type}}, \quad \mathbf{G}_{\text{removed}} = \sum_{\substack{\text{type} \in \\ \{\text{and,or}\}}} \sum_{S \in \Omega^{\text{type}}} |I_S^{\text{type}}| \cdot R_S^{\text{type}} \cdot G_S^{\text{type}} \tag{8}$$

In this way, let $\mathbf{A}_{\text{removed}}(r_1)$ and $\mathbf{A}_{\text{removed}}(r_2)$ denote the total strength of the removed interactions at two compression ratios $r_1$ and $r_2$ ($r_2 > r_1$). Correspondingly, we compute $\mathbf{G}_{\text{removed}}(r_1)$ and $\mathbf{G}_{\text{removed}}(r_2)$ based on the generalizable interactions. Then, the efficiency of parameter pruning $e^{r_1 \to r_2}$ within the interval $[r_1, r_2]$ is defined as follows.

$$e^{r_1 \to r_2} = 1 - \frac{\mathbf{G}_{\text{removed}}(r_2) - \mathbf{G}_{\text{removed}}(r_1)}{\mathbf{A}_{\text{removed}}(r_2) - \mathbf{A}_{\text{removed}}(r_1)} \tag{9}$$

This metric reflects the ratio of non-generalizable interactions to all the removed interactions. A higher value of $e^{r_1 \to r_2}$ indicates that less generalizable interactions are newly removed at pruning ratio $r_2$, compared with the DNN obtained with the pruning ratio $r_1$. We conducted experiments to illustrate the generalizability of the removed interactions after parameter pruning. According to Definition 2.2, we need to train reference DNNs from scratch on the testing samples. However, for large-scale models, it is often difficult to identify their training and testing samples. More importantly, retraining the model from scratch is impractical. Therefore, we used other classic models that were independently trained as the reference DNNs. Please see Appendix H for details.

Table 2 shows that parameter pruning exhibits fully different efficiency values $e^{r_1 \to r_2}$ in different phases. Specifically, **(1)** in the first phase when the pruning ratio increases is low, the pruning operation mainly removes high-order interactions, but few generalizable interaction are removed, thereby exhibiting a high pruning efficiency $e^{r_1 \to r_2}$. The performance of the DNN is not affected significantly, and generalizability of the entire DNN is largely preserved after parameter pruning. **(2)** In the second phase, both low-order interactions and high-order interactions are gradually removed. In this way, generalizable interactions (mainly of low orders) are removed, and the efficiency value $e^{r_1 \to r_2}$ begins to decreases. As a result, the testing accuracy of the DNN decreases significantly. **(3)** In the third phase, since most interactions in the original DNN have been removed, the performance of the DNN does not decrease a lot, and the very few removed interactions lead to an unstable efficiency $e^{r_1 \to r_2}$.

**Exploring generalizability of the emerged interactions during the three-phase dynamics.** In addition to removing interactions, we notice that the parameter-pruning operation may also bring in some new interactions. *Generally, we consider these newly emerged interactions to be purely noise patterns, because parameter pruning does not involve any data-driven learning process. In order to verify this hypothesis, we define and quantify the emerged interactions as follows.*

**Definition 2.4** *Let us be given a salient AND interaction $S$ extracted by the pruned DNN $v'(\cdot)$ from the input $\boldsymbol{x}$, subject to $|I'^{and}_S| > \tau$. If this interaction is negligible in the original DNN $v(\cdot)$ (subject to $|I^{and}_S| \le \tau$), then this interaction is considered as an emerged interaction after parameter pruning. The emergence of an OR interaction is defined similarly. Thus, we use the following binary metrics $E^{and}_S, E^{or}_S \in \{0, 1\}$ to represent the emergence of AND-OR interactions.*

$$E^{and}_S = \mathbb{1}(|I^{and}_S| \le \tau \ \text{and} \ |I'^{and}_S| > \tau), \quad E^{or}_S = \mathbb{1}(|I^{or}_S| \le \tau \ \text{and} \ |I'^{and}_S| > \tau) \tag{10}$$

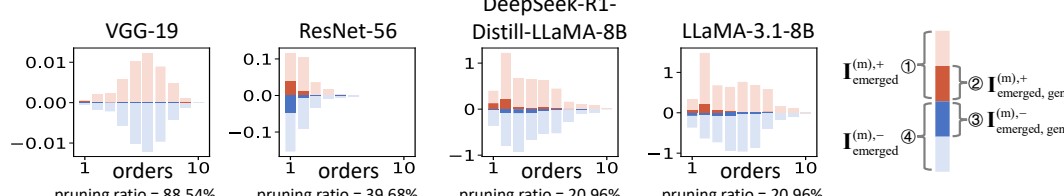

Figure 5: The distributions of $\mathbf{I}_{\text{emerged}}^{(m),+}$, $\mathbf{I}_{\text{emerged}}^{(m),-}$, $\mathbf{I}_{\text{emerged,gen}}^{(m),+}$ and $\mathbf{I}_{\text{emerged,gen}}^{(m),-}$. Emerged interactions are barely generalizable. **Please see Appendix J for results on more DNNs.**

In this way, the distribution of emerged interactions over different orders after parameter pruning can be quantified by the total strength of all positive and negative emerged interactions, denoted by $\mathbf{I}_{\text{emerged}}^{(m),+} = \mathbf{I}^{(m),+}(\Omega_{\text{emerged}}^{\text{and}}, \Omega_{\text{emerged}}^{\text{or}})$ and $\mathbf{I}_{\text{emerged}}^{(m),-} = \mathbf{I}^{(m),-}(\Omega_{\text{emerged}}^{\text{and}}, \Omega_{\text{emerged}}^{\text{or}})$, respectively, and the total strength of all positive and negative emerged generalizable interactions, denoted by $\mathbf{I}_{\text{emerged,gen}}^{(m),+} = \mathbf{I}^{(m),+}(\Omega_{\text{emerged,gen}}^{\text{and}}, \Omega_{\text{emerged,gen}}^{\text{or}})$ and $\mathbf{I}_{\text{emerged,gen}}^{(m),-} = \mathbf{I}^{(m),-}(\Omega_{\text{emerged,gen}}^{\text{and}}, \Omega_{\text{emerged,gen}}^{\text{or}})$, respectively.

$$
\begin{aligned}
\Omega_{\text{emerged}}^{\text{and}} &= \{S \in \Omega^{\text{and}} : E_S^{\text{and}} = 1\}, \quad \Omega_{\text{emerged}}^{\text{or}} = \{S \in \Omega^{\text{or}} : E_S^{\text{or}} = 1\} \\
\Omega_{\text{emerged,gen}}^{\text{and}} &= \{S \in \Omega^{\text{and}} : E_S^{\text{and}} = G_S^{\text{and}} = 1\}, \quad \Omega_{\text{emerged,gen}}^{\text{or}} = \{S \in \Omega^{\text{or}} : E_S^{\text{or}} = G_S^{\text{or}} = 1\}
\end{aligned}
\tag{11}
$$

where $\Omega_{\text{emerged}}^{\text{and}}$ and $\Omega_{\text{emerged}}^{\text{or}}$ denote the sets of emerged interactions, and $\Omega_{\text{emerged,gen}}^{\text{and}} \subseteq \Omega_{\text{emerged}}^{\text{and}}$ and $\Omega_{\text{emerged,gen}}^{\text{or}} \subseteq \Omega_{\text{emerged}}^{\text{or}}$ denote the subsets of interactions that are generalizable, respectively.

We conducted experiments to illustrate the generalizability of the emerged interactions after parameter pruning. We used the same reference DNN as in Section 2.3, when we analyzed generalizability of removed interactions. Figure 5 shows the distribution of the emerged interations brought by the parameter pruning operation at a certain pruning ratio. Most emerged interactions exhibts mutually offsetting effects and have low generalizability. A half of emerged interactions boost the classification score, and the other half decrease the classification score. According to He et al. (2025), such newly emerged mutually offsetting interactions usually represent fully noise patterns.

In summary, parameter pruning affects the DNN in two ways. (1) It removes both generalizable and non-generalizable interactions, and (2) it also brings in some new non-generalizable interactions. The observed alignment between the change of a DNN's generalizability and the dynamics of interactions further validates strong connection between the generalizability of the entire DNN and the generalizability of its compositional interactions.

# 3 CONCLUSION

In this paper, we have discovered a specific three-phase dynamics of interactions encoded by DNNs, when parameters are pruned under progressively increasing pruning ratios. Compared to traditional black-box evaluations, interaction-based analysis offers a more insightful perspective to understand the explicit influence of parameter pruning on the complexity and generalizability of a DNN's interaction representations. Specifically, we find that low-order interactions generally exhibit substantially stronger generalizability than high-order ones, and they tend to persist until relatively high pruning ratios are reached. Since we have discovered that performance degradation in neural networks can be largely attributed to the removal of generalizable low-order interactions, this work provides a new perspective to analyze the optimal compression ratio at the level of interactions.

While the three-phase dynamics of interactions has been consistently observed across multiple neural networks and tasks, a rigorous theoretical account of its root cause remains absent. In particular, it it is unclear why low-order interactions exhibit significantly greater robustness to pruning.

Our method is of considerable application value. In preliminary follow-up works, we penalize all non-generalizable interactions during the training of the DNN, which enables the DNN to be compressed at higher pruning ratios while maintaining its performance (see Appendix B).

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

## A RELATED WORK ON SYMBOLIC GENERALIZABILITY

**Using interactions to explain the detailed inference logic encoded by a DNN.** Prior works on symbolic generalizability (Kang et al., 2024; Li & Zhang, 2023b; Tsai et al., 2023; Sundararajan et al., 2020) proposed to explain the inference logic of a DNN by quantifying the interactions among input variables encoded by the DNN. Building on this, Ren et al. (2024a) further proved that networks producing relatively smooth outputs across different input perturbations typically encode a small number of interactions. Chen et al. (2024) developed an algorithm to extract interactions that were commonly encoded by multiple different neural networks. Ren et al. (2023b) further improved a method to optimize the baseline value, which was used for interaction extraction.

**Using interactions to explain the generalizability of a DNN.** Futhermore, symbolic generalizability research explain a DNN's generalizability as the overall generalizability of its compositional interactions. Zhou et al. (2024) found that high-order interactions usually generalize worse to testing samples than lower-order interactions. citetren2021towards further found that these higher-order interactions also showed poorer adversarial robustness. Ren et al. (2023c) found that mean-field Bayesian neural networks typically struggled more than regular neural networks in modeling higher-order interactions.

Although symbolic generalizability offers a powerful strategy to explain the inference logic of DNNs and their generalizability across different data domains, existing studies still leave a blank in the

field of post-training interventional techniques. To this end, by investigating parameter pruning techniques, this paper bridges a crucial gap between symbolic generalizability theory and parameter pruning techniques, and offers a new perspective for understanding how interventional techniques, such as parameter pruning operations, reshape network representation quality and generalizability.

**Comparison with mechanistic interpretability.** Mechanistic interpretability treats a neural network as a collection of distinct components rather than a single black box (Zhou et al., 2019; Meng et al., 2022; Wang et al., 2023). It studies the roles of individual neurons, attention heads, or sub-networks, and how these parts work together to produce the model's overall behavior. Instead of physically dividing a neural network into sub-components, symbolic generalizability treats a neural network as a whole and explains the inference logic equivalently encoded by the network using a set of AND-OR interactions.

## B  PRACTICAL VALUE OF OUR FINDINGS

According to our findings, low-order generalizable interactions are removed at higher pruning ratios than high-order non-generalizable interactions. Based on this observation, in follow-up studies we penalize the strength of all non-generalizable interactions during DNN training. This method encourages the DNN to encode more generalizable interactions.

We trained VGG-16 (Simonyan & Zisserman, 2015) and AlexNet (Krizhevsky et al., 2012) on the TinyImageNet dataset (Le & Yang, 2015), both with and without the proposed penalty.

As shown in Figure 6, DNNs trained with this penalty maintained their performance under higher pruning ratios compared to DNNs trained without it.

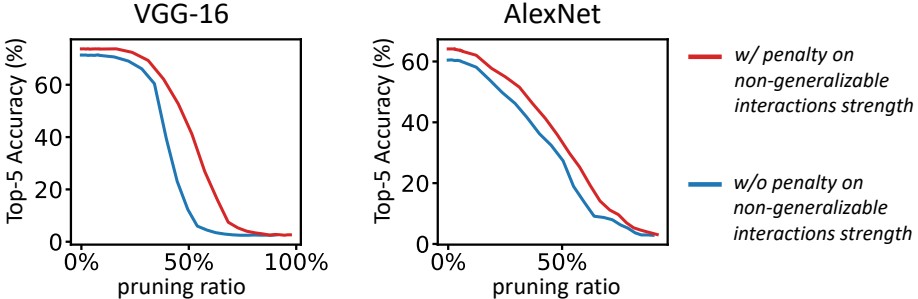

Figure 6: Top-5 classification accuracy of pruned VGG-16 and pruned AlexNet on TinyImageNet under different pruning ratio.

## C  PROPERTIES OF THE AND INTERACTION

The Harsanyi interaction Harsanyi (1963) (referred to as the AND interaction in this work) has been a conventional metric for measureing the effect of the AND relationship that a DNN encodes among input variables. In this section, we introduce several desirable axioms that the AND interaction $I_T^{\text{and}}$ adheres to. These properties further underscore the reliability of using AND interactions to explain the inference score of a DNN.

(1) *Efficiency axiom* (proven by Harsanyi (1963)). The output score of a model can be decomposed into interaction effects of different patterns, *i.e.* $v(\boldsymbol{x}) = \sum_{T \subseteq N} I_T^{\text{and}}$.

(2) *Linearity axiom.* If we merge output scores of two models $v_1$ and $v_2$ as the output of model $v$, *i.e.* $\forall S \subseteq N$, $v(\boldsymbol{x}_S) = v_1(\boldsymbol{x}_S) + v_2(\boldsymbol{x}_S)$, then their interaction effects $I_{T,v_1}^{\text{and}}$ and $I_{T,v_2}^{\text{and}}$ can also be merged as $\forall T \subseteq N, I_{T,v}^{\text{and}} = I_{T,v_1}^{\text{and}} + I_{T,v_2}^{\text{and}}$.

(3) *Dummy axiom.* If a variable $i \in N$ is a dummy variable, *i.e.* $\forall S \subseteq N \setminus \{i\}, v(\boldsymbol{x}_{S\cup\{i\}}) = v(\boldsymbol{x}_S) + v(\boldsymbol{x}_{\{i\}})$, then it has no interaction with other variables, $\forall \emptyset \neq T \subseteq N \setminus \{i\}, I_{T\cup\{i\}}^{\text{and}} = 0$.

(4) *Symmetry axiom.* If input variables $i, j \in N$ cooperate with other variables in the same way, $\forall S \subseteq N \setminus \{i, j\}, v(\boldsymbol{x}_{S \cup \{i\}}) = v(\boldsymbol{x}_{S \cup \{j\}})$, then they have same interaction effects with other variables, $\forall T \subseteq N \setminus \{i, j\}, I_{T \cup \{i\}}^{\text{and}} = I_{T \cup \{j\}}^{\text{and}}$.

(5) *Anonymity axiom.* For any permutations $\pi$ on $N$, we have $\forall T \subseteq N, I_{T,v}^{\text{and}} = I_{\pi T, \pi v}^{\text{and}}$, where $\pi T \overset{\text{def}}{=} \{\pi(i) | i \in T\}$, and the new model $\pi v$ is defined by $(\pi v)(\boldsymbol{x}_{\pi S}) = v(\boldsymbol{x}_S)$. This indicates that interaction effects are not changed by permutation.

(6) *Recursive axiom.* The interaction effects can be computed recursively. For $i \in N$ and $T \subseteq N \setminus \{i\}$, the interaction effect of the pattern $T \cup \{i\}$ is equal to the interaction effect of $T$ with the presence of $i$ minus the interaction effect of $T$ with the absence of $i$, *i.e.* $\forall T \subseteq N \setminus \{i\}, I_{T \cup \{i\}}^{\text{and}} = I_{T, i \text{ present}}^{\text{and}} - I_T^{\text{and}}$. $I_{T, i \text{ present}}^{\text{and}}$ denotes the interaction effect when the variable $i$ is always present as a constant context, *i.e.* $I_{T, i \text{ present}}^{\text{and}} = \sum_{L \subseteq T} (-1)^{|T| - |L|} \cdot v(\boldsymbol{x}_{L \cup \{i\}})$.

(7) *Interaction distribution axiom.* This axiom characterizes how interactions are distributed for "interaction functions" Sundararajan et al. (2020). An interaction function $v_T$ parameterized by a subset of variables $T$ is defined as follows. $\forall S \subseteq N$, if $T \subseteq S, v_T(\boldsymbol{x}_S) = c$ ; otherwise, $v_T(\boldsymbol{x}_S) = 0$. The function $v_T$ models pure interaction among the variables in $T$, because only if all variables in $T$ are present, the output value will be increased by $c$. The interactions encoded in the function $v_T$ satisfies $I_T^{\text{and}} = c$, and $\forall S \neq T, I_S^{\text{and}} = 0$.

# D  DIFFERENT PROPERTIES OF LOW-ORDER AND HIGH-ORDER INTERACTIONS

As Figure 3 and Figure 26 show, interactions of different orders exhibit distinct properties.

(1) Among all low-order interactions (including interactions of the $1st$ and $3rd$ orders), there are more low-order interactions with positive effects than those with negative effects, suggesting that low-order interactions serve as the primary inference patterns to boost the classification confidence. (2) Most high-order interactions, which range from the $4th$ to the $n$-$th$ order, usually exhibit mutually offsetting effects, *i.e.*, a half of the high-order interactions increase the classification confidence, while the other half decrease the classification confidence.

# E  COMMON CONDITIONS FOR SPARSE INTERACTIONS

Ren et al. (2024a) have proved three sufficient conditions for the sparsity of AND interactions.

**Condition 1.** *The DNN does not encode extremely high-order interactions:* $\forall \, T \in \{T \subseteq N \mid |T| \geq M + 1\}, \, I_T^{\text{and}} = 0$.

Condition 1 is common because extremely high-order interactions usually represent very complex and over-fitted patterns, which are unlikely to be learned by a well-trained DNN in real scenarios.

**Condition 2.** *Let $\bar{u}^{(k)} \overset{\text{def}}{=} \mathbb{E}_{|S|=k}[v(\boldsymbol{x}_S) - v(\boldsymbol{x}_\emptyset)]$ denote the average classification confidence of the DNN over all masked samples $\boldsymbol{x}_S$ with $k$ unmasked input variables. This average classification confidence monotonically increases when $k$ increases:* $\forall \, k' \leq k, \, \bar{u}^{(k')} \leq \bar{u}^{(k)}$.

Condition 2 implies that a well-trained DNN is likely to have higher average classification confidence for less masked input samples.

**Condition 3.** *Given the average classification confidence $\bar{u}^{(k)}$ of samples with $k$ unmasked input variables, there is a polynomial lower bound for the average classification confidence with $k'(k' \leq k)$ unmasked input variables:* $\forall \, k' \leq k, \, \bar{u}^{(k')} \geq (\frac{k'}{k})^p \, \bar{u}^{(k)}$, *where $p > 0$ is a constant.*

Condition 3 suggests that the classification confidence of the DNN remains relatively stable even when presented with masked input samples. In real-world applications, the classification or detection of masked or occluded samples frequently occurs. As a result, a well-trained DNN typically develops the ability to classify such masked inputs by leveraging local information, which can be derived from the visible portions of the input. Consequently, the model should not produce a substantially reduced confidence score for masked samples.

# F    DETAILS OF EXTRACTING THE SPARSEST AND-OR INTERACTIONS

A method is proposed Chen et al. (2024); Li & Zhang (2023a) to simultaneously extract AND interactions $I_T^{\text{and}}$ and OR interactions $I_T^{\text{OR}}$ from the network output. Given a masked sample $\boldsymbol{x}_L$, Li & Zhang (2023a) proposed to learn a decomposition $v(\boldsymbol{x}_L) = u_L^{\text{and}} + u_L^{\text{OR}}$ towards the sparsest interactions. The component $u_L^{\text{and}}$ was explained by AND interactions, and the component $u_L^{\text{OR}}$ was explained by OR interactions. Specifically, they decomposed $v(\boldsymbol{x}_L)$ into $u_L^{\text{and}} = 0.5 \cdot v(\boldsymbol{x}_L) + \gamma_L$ and $u_L^{\text{OR}} = 0.5 \cdot v(\boldsymbol{x}_L) - \gamma_L$, where $\{\gamma_L : L \subseteq N\}$ is a set of learnable variables that determine the decomposition. In this way, the AND interactions and OR interactions can be computed according to Theorem 2.1, *i.e.*, $I_T^{\text{and}} = \sum_{L \subseteq T}(-1)^{|T|-|L|} u_L^{\text{and}}$, and $I_T^{\text{OR}} = -\sum_{L \subseteq T}(-1)^{|T|-|L|} u_{N \setminus L}^{\text{OR}}$.

The parameters $\{\gamma_L\}$ were learned by minimizing the following LASSO-like loss to obtain sparse interactions:

$$\min_{\{\gamma_L\}} \sum_{T \subseteq N} |I_T^{\text{and}}| + |I_T^{\text{OR}}| \tag{12}$$

The following pseudocode 1 outlines the core procedure of extracting AND-OR interactions.

---

**Algorithm 1** Computing AND-OR interactions

---

1: **Input:** Input sample $\boldsymbol{x}$, the DNN $v(\cdot)$
2: **Output:** A set of interactions $I_S^{\text{AND}}$ and $I_S^{\text{OR}}$
3: **for** $S \subseteq N$ **do**
4:     For each masked sample $\boldsymbol{x}_S$, compute the confidence score $v(\boldsymbol{x}_S)$ based on Eq. (1);
5: **end for**
6: **for** $S \subseteq N$ **do**
7:     Given $v(\boldsymbol{x}_S)$ for all combinations $S \subseteq N$, compute each AND interaction effect $I_S^{\text{AND}}$ and each OR interaction effect $I_S^{\text{OR}}$ via $\min_{\{\gamma_T\}} \sum_{S \subseteq N, S \neq \emptyset}[|I_S^{\text{AND}}| + |I_S^{\text{OR}}|]$;
8: **end for**
9: return $I_S^{\text{AND}}, I_S^{\text{OR}}$

---

Furthermore, to extract interactions shared between the original DNN $v(\boldsymbol{x})$ and the pruned DNN $v'(\boldsymbol{x})$, we modified the objective as:

$$\min_{\{\gamma_L\}} \quad \sum_{T \subseteq N} \left( |I_T^{\text{and}}| + |I_T^{\text{or}}| + |I_T'^{\text{and}}| + |I_T'^{\text{or}}| \right)$$

$$+ \sum_{T \subseteq N} \left( |\min(0, I_T^{\text{and}}, I_T'^{\text{and}})| + |\max(0, I_T^{\text{and}}, I_T'^{\text{and}})| \right. \tag{13}$$

$$\left. + |\min(0, I_T^{\text{or}}, I_T'^{\text{or}})| + |\max(0, I_T^{\text{or}}, I_T'^{\text{or}})| \right)$$

**Removing small noises.** A small noise $\delta$ in the network output may significantly affect the extracted interactions, especially for high-order interactions. Thus, Li et al. Li & Zhang (2023a) proposed to learn to remove a small noise term $\delta_T$ from the computation of AND-OR interactions. Specifically, the decomposition was rewritten as $u_L^{\text{and}} = 0.5(v(\boldsymbol{x}_L) - \delta_L) + \gamma_L$ and $u_L^{\text{OR}} = 0.5(v(\boldsymbol{x}_L) - \delta_L) + \gamma_L$. Thus, the parameters $\{\delta_L\}$ and $\{\gamma_L\}$ are simultaneously learned by minimizing the loss function in Eq. (12). The values of $\{\delta_L\}$ were constrained in $[-\zeta, \zeta]$ where $\zeta = 0.02 \cdot |v(\boldsymbol{x}) - v(\boldsymbol{x}_\emptyset)|$.

# G    PROOF OF THEOREM 2.1

**Proof G.1** *(1) Universal matching property of AND interactions.*

*We will prove that output component $u_S^{AND}$ on all $2^n$ masked samples $\{\boldsymbol{x}_S : S \subseteq N\}$ could be universally explained by the all interactions in $S \subseteq N$, i.e., $\forall \emptyset \neq S \subseteq N, u_S^{AND} = \sum_{\emptyset \neq T \subseteq S} I_T^{AND} + v(\boldsymbol{x}_\emptyset)$. In particular, we define $u_\emptyset^{AND} = v(\boldsymbol{x}_\emptyset)$ (i.e., we attribute output on an empty sample to AND interactions).*

*Specifically, the AND interaction is defined as $I_T^{AND} = \sum_{L \subseteq T} (-1)^{|T|-|L|} u_L^{AND}$. To compute the sum of AND interactions $\sum_{\emptyset \neq T \subseteq S} I_T^{AND} = \sum_{\emptyset \neq T \subseteq S} \sum_{L \subseteq T} (-1)^{|T|-|L|} u_L^{AND}$, we first exchange the order of summation of the set $L \subseteq T \subseteq S$ and the set $T \supseteq L$. That is, we compute all linear combinations of all sets $T$ containing $L$ with respect to the model outputs $u_L^{AND}$ given a set of input variables $L$, i.e., $\sum_{T:L \subseteq T \subseteq S} (-1)^{|T|-|L|} u_L^{AND}$. Then, we compute all summations over the set $L \subseteq S$.*

*In this way, we can compute them separately for different cases of $L \subseteq T \subseteq S$. In the following, we consider the cases (1) $L = S = T$, and (2) $L \subseteq T \subseteq S, L \neq S$, respectively.*

*(1) When $L = S = T$, the linear combination of all subsets $T$ containing $L$ with respect to the model output $u_L^{AND}$ is $(-1)^{|S|-|S|} u_L^{AND} = u_L^{AND}$.*

*(2) When $L \subseteq T \subseteq S, L \neq S$, the linear combination of all subsets $T$ containing $L$ with respect to the model output $u_L^{AND}$ is $\sum_{T:L \subseteq T \subseteq S} (-1)^{|T|-|L|} u_L^{AND}$. For all sets $T : S \supseteq T \supseteq L$, let us consider the linear combinations of all sets $T$ with number $|T|$ for the model output $u_L^{AND}$, respectively. Let $m := |T| - |L|$, $(0 \leq m \leq |S| - |L|)$, then there are a total of $C_{|S|-|L|}^m$ combinations of all sets $T$ of order $|T|$. Thus, given $L$, accumulating the model outputs $u_L^{AND}$ corresponding to all $T \supseteq L$, then $\sum_{T:L \subseteq T \subseteq S} (-1)^{|T|-|L|} u_L^{AND} = u_L^{AND} \cdot \underbrace{\sum_{m=0}^{|S|-|L|} C_{|S|-|L|}^m (-1)^m}_{=0} = 0$. Please see the complete derivation of the following formula.*

$$
\begin{aligned}
\sum_{\emptyset \neq T \subseteq S} I_T^{AND} &= \sum_{\emptyset \neq T \subseteq S} \sum_{L \subseteq T} (-1)^{|T|-|L|} u_L^{AND} \\
&= \sum_{L \subseteq S} \sum_{T:L \subseteq T \subseteq S} (-1)^{|T|-|L|} u_L^{AND} - u_\emptyset^{AND} \\
&= \underbrace{u_S^{AND}}_{L=S} + \sum_{L \subseteq S, L \neq S} u_L^{AND} \cdot \underbrace{\sum_{m=0}^{|S|-|L|} C_{|S|-|L|}^m (-1)^m}_{=0} - u_\emptyset^{AND} \\
&= u_S^{AND} - u_\emptyset^{AND} = u_S^{AND} - v(\boldsymbol{x}_\emptyset)
\end{aligned}
\tag{14}
$$

*Thus, we have $\forall \emptyset \neq S \subseteq N, u_S^{and} = \sum_{\emptyset \neq T \subseteq S} I_T^{and} + v(\boldsymbol{x}_\emptyset)$.*

**(2) Universal matching property of OR interactions.**

*According to the definition of OR interactions, we will derive that $\forall S \subseteq N, u_S^{OR} = \sum_{T:T \cap S \neq \emptyset} I_T^{OR}$, where we define $u_\emptyset^{OR} = 0$ (recall that in Step (1), we attribute the output on empty input to AND interactions).*

*Specifically, the OR interaction is defined as $I_T^{OR} = -\sum_{L \subseteq T} (-1)^{|T|-|L|} u_{N \setminus L}^{OR}$. Similar to the above derivation of the universal matching theorem of AND interactions, to compute the sum of OR interactions $\sum_{T:T \cap S \neq \emptyset} I_T^{OR} = \sum_{T:T \cap S \neq \emptyset} \left[ -\sum_{L \subseteq T} (-1)^{|T|-|L|} u_{N \setminus L}^{OR} \right]$, we first exchange the order of summation of the set $L \subseteq T \subseteq N$ and the set $T : T \cap S \neq \emptyset$. That is, we compute all linear combinations of all sets $T$ containing $L$ with respect to the model outputs $u_{N \setminus L}^{OR}$ given a set of input variables $L$, i.e., $\sum_{T:T \cap S \neq \emptyset, T \supseteq L} (-1)^{|T|-|L|} u_{N \setminus L}^{OR}$. Then, we compute all summations over the set $L \subseteq N$.*

*In this way, we can compute them separately for different cases of $L \subseteq T \subseteq N, T \cap S \neq \emptyset$. In the following, we consider the cases (1) $L = N \setminus S$, (2) $L = N$, (3) $L \cap S \neq \emptyset, L \neq N$, and (4) $L \cap S = \emptyset, L \neq N \setminus S$, respectively.*

*(1) When $L = N \setminus S$, the linear combination of all subsets $T$ containing $L$ with respect to the model output $u_{N \setminus L}^{OR}$ is $\sum_{T:T \cap S \neq \emptyset, T \supseteq L} (-1)^{|T|-|L|} u_{N \setminus L}^{OR} = \sum_{T:T \cap S \neq \emptyset, T \supseteq L} (-1)^{|T|-|L|} u_S^{OR}$. For all sets $T : T \supseteq L, T \cap S \neq \emptyset$ (then $T \neq N \setminus S, T \neq L$), let us consider the linear combinations of all sets $T$ with number $|T|$ for the model output $u_S^{OR}$, respectively. Let $|T'| := |T| - |L|$, $(1 \leq |T'| \leq |S|)$, then there are a total of $C_{|S|}^{|T'|}$ combinations of all sets $T'$ of order $|T'|$. Thus, given $L$, accumulating*

the model outputs $u_S^{OR}$ corresponding to all $T \supseteq L$, then $\sum_{T:T\cap S\neq\emptyset, T\supseteq L}(-1)^{|T|-|L|}u_{N\setminus L}^{OR} = u_S^{OR} \cdot$

$$\underbrace{\sum_{|T'|=1}^{|S|}C_{|S|}^{|T'|}(-1)^{|T'|}}_{=-1} = -u_S^{OR}.$$

*(2) When $L = N$ (then $T = N$), the linear combination of all subsets $T$ containing $L$ with respect to the model output $u_{N\setminus L}^{OR}$ is $\sum_{T:T\cap S\neq\emptyset, T\supseteq L}(-1)^{|T|-|L|}u_{N\setminus L}^{OR} = (-1)^{|N|-|N|}u_\emptyset^{OR} = u_\emptyset^{OR}$.*

*(3) When $L\cap S\neq\emptyset, L\neq N$, the linear combination of all subsets $T$ containing $L$ with respect to the model output $u_{N\setminus L}^{OR}$ is $\sum_{T:T\cap S\neq\emptyset, T\supseteq L}(-1)^{|T|-|L|}u_{N\setminus L}^{OR}$. For all sets $T : T \supseteq L, T\cap S\neq\emptyset$, let us consider the linear combinations of all sets $T$ with number $|T|$ for the model output $u_S^{OR}$, respectively. Let us split $|T| - |L|$ into $|T'|$ and $|T''|$, i.e., $|T| - |L| = |T'| + |T''|$, where $T' = \{i|i\in T, i\notin L, i\in N\setminus S\}$, $T'' = \{i|i\in T, i\notin L, i\in S\}$ (then $0\le|T''|\le|S|-|S\cap L|$) and $|T'| + |T''| + |L| = |T|$. In this way, there are a total of $C_{|S|-|S\cap L|}^{|T''|}$ combinations of all sets $T''$ of order $|T''|$. Thus, given $L$, accumulating the model outputs $u_{N\setminus L}^{OR}$ corresponding to all $T \supseteq L$, then*

$$\sum_{T:T\cap S\neq\emptyset, T\supseteq L}(-1)^{|T|-|L|}u_{N\setminus L}^{OR} = u_{N\setminus L}^{OR}\cdot\sum_{T'\subseteq N\setminus S\setminus L}\underbrace{\sum_{|T''|=0}^{|S|-|S\cap L|}C_{|S|-|S\cap L|}^{|T''|}(-1)^{|T'|+|T''|}}_{=0} =$$

0.

*(4) When $L\cap S = \emptyset, L\neq N\setminus S$, the linear combination of all subsets $T$ containing $L$ with respect to the model output $u_{N\setminus L}^{OR}$ is $\sum_{T:T\cap S\neq\emptyset, T\supseteq L}(-1)^{|T|-|L|}u_{N\setminus L}^{OR}$. Similarly, let us split $|T| - |L|$ into $|T'|$ and $|T''|$, i.e., $|T| - |L| = |T'| + |T''|$, where $T' = \{i|i\in T, i\notin L, i\in N\setminus S\}$, $T'' = \{i|i\in T, i\in S\}$ (then $0 \le |T''| \le |S|$) and $|T'| + |T''| + |L| = |T|$. In this way, there are a total of $C_{|S|}^{|T''|}$ combinations of all sets $T''$ of order $|T''|$. Thus, given $L$, accumulating the model outputs $u_{N\setminus L}^{OR}$ corresponding to all $T \supseteq L$, then $\sum_{T:T\cap S\neq\emptyset, T\supseteq L}(-1)^{|T|-|L|}u_{N\setminus L}^{OR} =$*

$$u_{N\setminus L}^{OR}\cdot\sum_{T'\subseteq N\setminus S\setminus L}\underbrace{\sum_{|T''|=0}^{|S|}C_{|S|}^{|T''|}(-1)^{|T'|+|T''|}}_{=0} = 0.$$

*Please see the complete derivation of the following formula.*

$$\begin{aligned}
\sum_{T:T\cap S\neq\emptyset}I_T^{OR} &= \sum_{T:T\cap S\neq\emptyset}\left[-\sum_{L\subseteq T}(-1)^{|T|-|L|}u_{N\setminus L}^{OR}\right] \\
&= -\sum_{L\subseteq N}\sum_{T:T\cap S\neq\emptyset, T\supseteq L}(-1)^{|T|-|L|}u_{N\setminus L}^{OR} \\
&= -\left[\sum_{|T'|=1}^{|S|}C_{|S|}^{|T'|}(-1)^{|T'|}\right]\cdot\underbrace{u_S^{OR}}_{L=N\setminus S} - \underbrace{u_\emptyset^{OR}}_{L=N} \\
&\quad - \sum_{L\cap S\neq\emptyset, L\neq N}\left[\sum_{T'\subseteq N\setminus S\setminus L}\left(\sum_{|T''|=0}^{|S|-|S\cap L|}C_{|S|-|S\cap L|}^{|T''|}(-1)^{|T'|+|T''|}\right)\right]\cdot u_{N\setminus L}^{OR} \\
&\quad - \sum_{L\cap S=\emptyset, L\neq N\setminus S}\left[\sum_{T'\subseteq N\setminus S\setminus L}\left(\sum_{|T''|=0}^{|S|}C_{|S|}^{|T''|}(-1)^{|T'|+|T''|}\right)\right]\cdot u_{N\setminus L}^{OR} \\
&= -(-1)\cdot u_S^{OR} - u_\emptyset^{OR} - \sum_{L\cap S\neq\emptyset, L\neq N}\left[\sum_{T'\subseteq N\setminus S\setminus L}0\right]\cdot u_{N\setminus L}^{OR} \\
&\quad - \sum_{L\cap S=\emptyset, L\neq N\setminus S}\left[\sum_{T'\subseteq N\setminus S\setminus L}0\right]\cdot u_{N\setminus L}^{OR} \\
&= u_S^{OR} - u_\emptyset^{OR} \\
&= u_S^{OR}
\end{aligned} \tag{15}$$

*(3) Universal matching property of AND-OR interactions.*

*With the universal matching theorem of AND interactions and the universal matching theorem of OR interactions, we can easily get $v(\boldsymbol{x}_S) = u_S^{and} + u_S^{OR} = v(\boldsymbol{x}_\emptyset) + \sum_{\emptyset \neq T \subseteq S} I_T^{and} + \sum_{T:T \cap S \neq \emptyset} I_T^{OR}$, thus, we obtain the universal matching theorem of AND-OR interactions.*

# H EXPERIMENTAL DETAIL

## H.1 TRAINING SETTINGS

In this paper, we followed Fang et al. (2023) and trained VGG-19 Simonyan & Zisserman (2015) and ResNet-56 He et al. (2016) on the CIFAR-10 dataset Krizhevsky (2009) for 200 epochs using stochastic gradient descent (SGD) with a learning rate of $0.1$, momentum of $0.9$, and a weight decay of $5 \times 10^{-4}$. The learning rate was decayed by a factor of $0.1$ at epochs 120, 150, and 180. We used a batch size of 128 in all experiments.

All experiments were conducted on a compute node equipped with dual Intel Xeon Silver 4310 CPUs (48 logical cores) and a combination of four NVIDIA A800 and two NVIDIA A100 GPUs (each with 80GB memory).

## H.2 PRUNING SETTINGS

Considering the adaptability of pruning algorithms to models, (1) we applied LLM-Pruner (Ma et al., 2023) to prune DeepSeek-R1-Distill-LLaMA-8B model (DeepSeek-AI et al., 2025) and LLaMA-3.1-8B model (Grattafiori et al., 2024); (2) we applied Isomorphic-Pruning (Fang et al., 2024) to prune ResNet-50, ResNet-101 (He et al., 2016; Wightman et al., 2021), ViT-Small and ViT-Base (Dosovitskiy et al., 2021; Steiner et al., 2022) pretrained on the ImageNet-1k dataset (Russakovsky et al., 2015); (3) we used DepGraph (Fang et al., 2023) to prune VGG-19, ResNet-56 on the CIFAR-10 dataset (Krizhevsky, 2009).

All other pruning settings followed the original settings in Ma et al. (2023); Fang et al. (2024; 2023), respectively. No post-pruning finetuning was performed.

## H.3 DETAILS ABOUT HOW TO CALCULATE INTERACTIONS FOR DIFFERENT DNNS

• **For experiments on image classification models,** since the computational cost of interactions was intolerable, we applied a sampling-based approximation method to calculate AND-OR interactions. Specifically, we considered the feature map after the low-layer as intermediate-layer features of DNNs. We uniformly split each intermediate-layer feature map into $s \times s$ patches, and sampled 10 patches based on the highest significance values computed via gradient integration. For VGG-19 and ResNet-56 on CIFAR-10, $s = 8$. For ResNet-50, ResNet-50, ViT-Small and ViT-Base on ImageNet-1k, $s = 7$. These selected patches were considered as input variables for the corresponding intermediate-layer feature. We used a specific feature vector as the baseline value to mask the variables in $N \backslash T$. The duration of the experiments ranges from 4 to 8 hours.

• **For experiments on large language models,** we manually collected 100 sentences for interaction extraction, which cover topics such as corporate restructuring, science and technology, economic policy, global trade, geopolitics, public health, political developments, cryptocurrency collapse, monetary policy, and climate and energy equity. The zero-shot perplexity (PPL) analysis is performed on WikiText2 dataset (Merity et al., 2017). We considered the outputs of the low-layer corresponding to input words as input features. We considered the embeddings corresponding to input features as input variables for each input sentence, and we randomly sampled 10 words, which must have a specific meaning and not be stop words, to calculate interactions. We used the average embedding over different input variables to mask the tokens in $N \backslash T$. We used a specific feature vector as the baseline value to mask the variables in $N \backslash T$. The duration of the experiments ranges from 6 to 8 hours.

Specifically, we empirically considered the first 4 convolutional layer of VGG-19 as the low layers, and considered all the other 13 layers[3] as the high layers. For ResNet-56, we consider the first 3

---

[3]VGG-19 for CIFAR-10 Krizhevsky (2009) has been simplified compared to the original VGG-19 architecture designed for ImageNet, by replacing the multi-layer perceptron (MLP) classifier with a single fully

convolution layers as low layers. For ResNet-50 and ResNet-101, we consider the first 11 convolution layers as low layers. For AlexNet, ViT-Small and ViT-Base, we consider the first convolution layer as low layers. Typically, we compute the mean distribution of interactions over 100 samples for ResNet-56 on the CIFAR-10 dataset. We compute the mean distribution of interactions over 100 samples for VGG-19 on the CIFAR-10 dataset. We compute the mean distribution of interactions over 100 samples for DeepSeek-R1-Distill-LLaMA-8B on the above manually collected data.

## H.4 Settings of reference DNNs

We used VGG-19 and ResNet-56 as each other's reference model. For ResNet-50 and ResNet-101, we used the ViT-Base model as reference DNN. For ViT-Small and ViT-Base, we used the ResNet-101 model as reference DNN. We used the Qwen-2.5-7B (Qwen et al., 2024) as the reference DNN for the DeepSeek-R1-Distill-LLaMA-8B and LLaMA-3.1-8B model.

## I Illustration of And-Or logical model

The following four figures show the logical models that mathematically represent the inference logic of DeepSeek-R1-Distill-LLaMA-8B model and Qwen-2.5-7B on two input prompts, respectively.

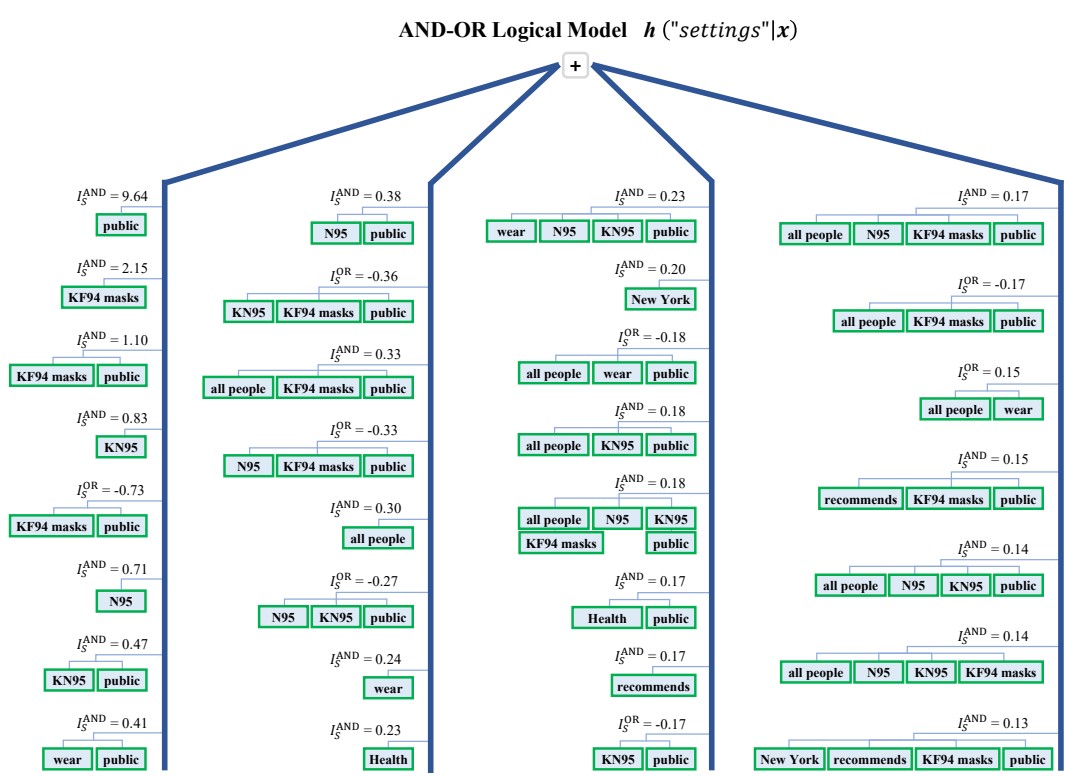

Figure 7: The logical model representing the inference logic of the DeepSeek-R1-Distill-LLaMA-8B model on the input prompt "New York Department of Health recommends that all people should wear N95, KN95, or KF94 masks in all public." The predicted next word is "settings."

connected layer and adapting the input resolution. The total number of layers, including both convolutional and fully connected layers, remains 17 in the simplified VGG-19 model for CIFAR-10.

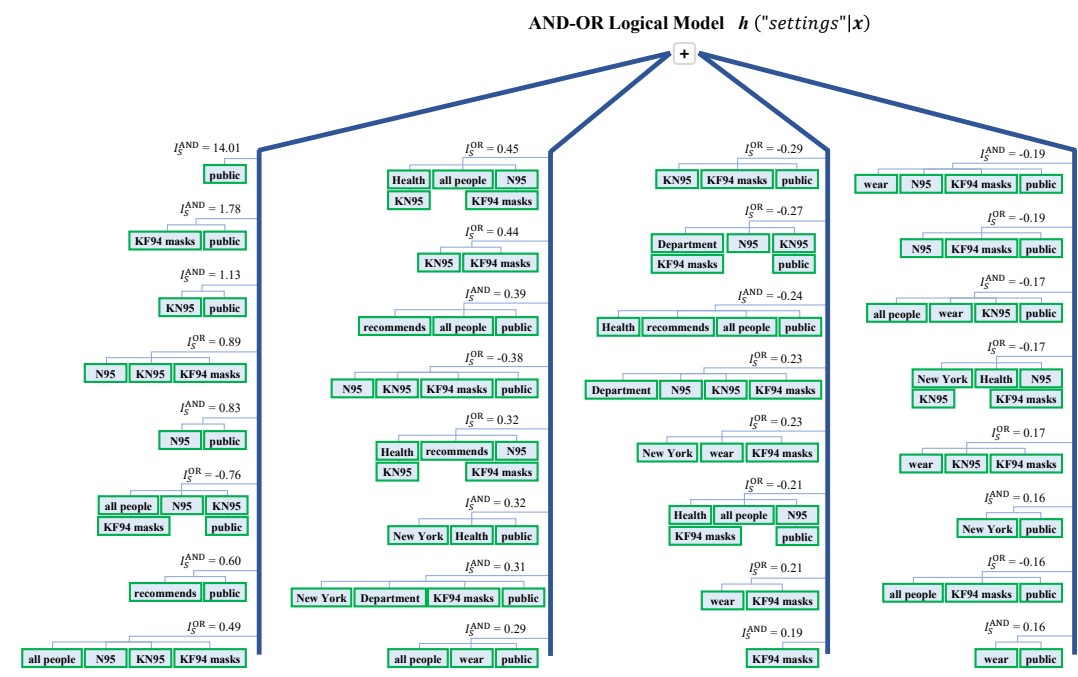

Figure 8: The logical model representing the inference logic of the Qwen-2.5-7B model on the input prompt "New York Department of Health recommends that all people should wear N95, KN95, or KF94 masks in all public." The predicted next word is "settings."

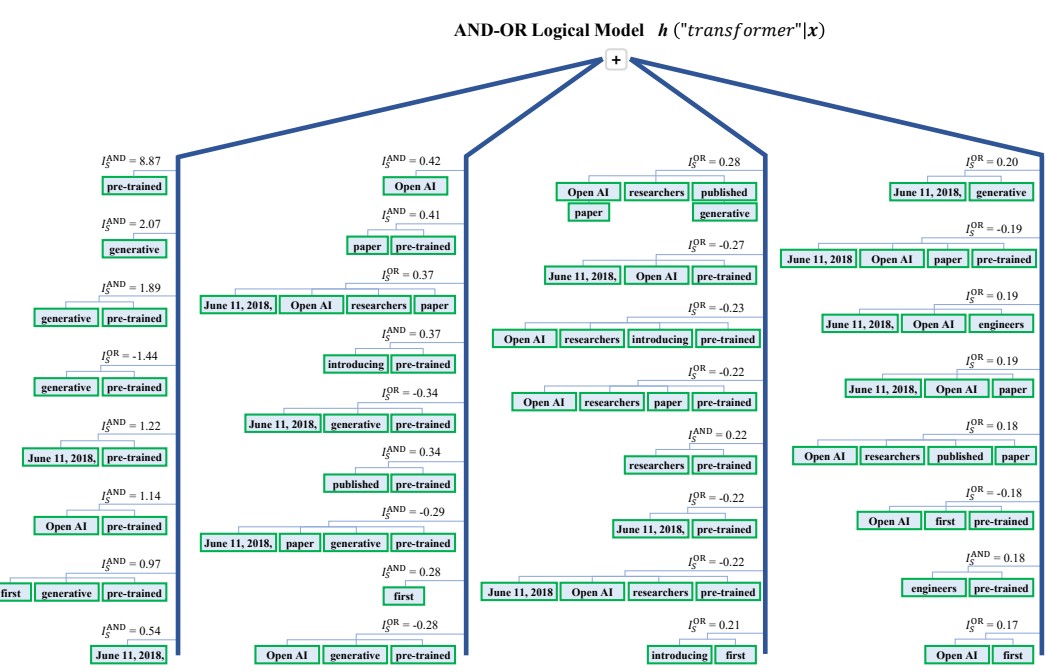

Figure 9: The logical model representing the inference logic of the DeepSeek-R1-Distill-LLaMA-8B model on the input prompt "On June 11, 2018, OpenAI researchers and engineers published a paper introducing the first generative pre-trained." The predicted next word is "transformer."

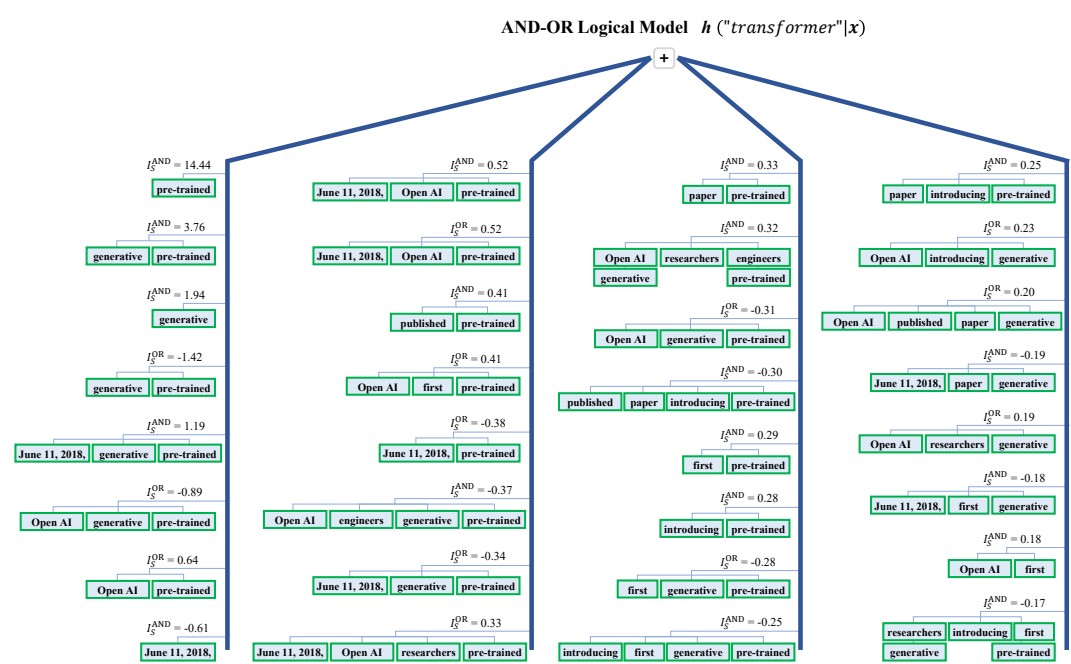

Figure 10: The logical model representing the inference logic of the Qwen-2.5-7B model on the input prompt "On June 11, 2018, OpenAI researchers and engineers published a paper introducing the first generative pre-trained." The predicted next word is "transformer."

The following four figures show the logical models that mathematically represent the inference logic of DeepSeek-R1-Distill-LLaMA-8B model (before and after pruning operation) on two input prompts, respectively.

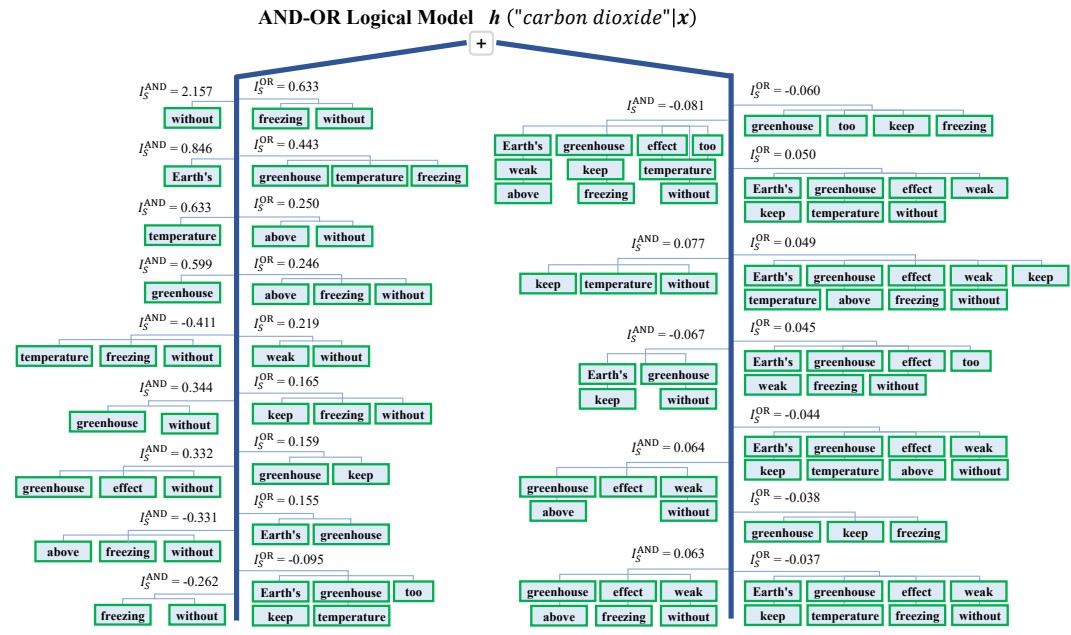

Figure 11: The logical model representing the inference logic of the original DeepSeek-R1-Distill-LLaMA-8B model on the input prompt "Earth's greenhouse effect would be too weak to keep temperature above freezing without." The predicted next word is "carbon dioxide."

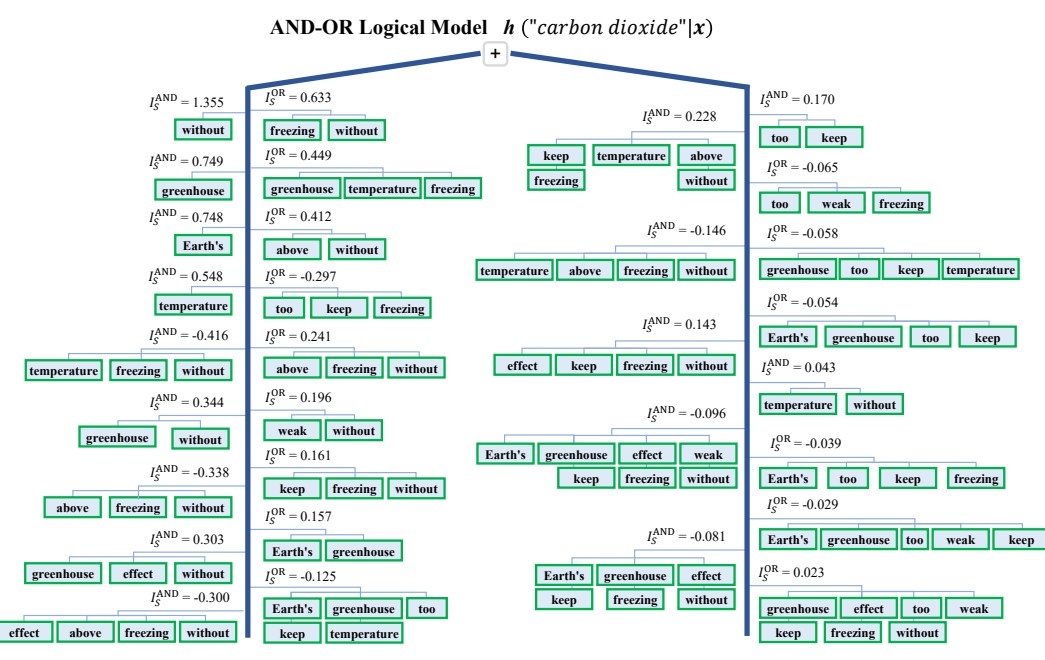

Figure 12: The logical model representing the inference logic of the pruned DeepSeek-R1-Distill-LLaMA-8B model (under a pruning ratio of 41.01) on the input prompt "Earth's greenhouse effect would be too weak to keep temperature above freezing without." The predicted next word is "carbon dioxide."

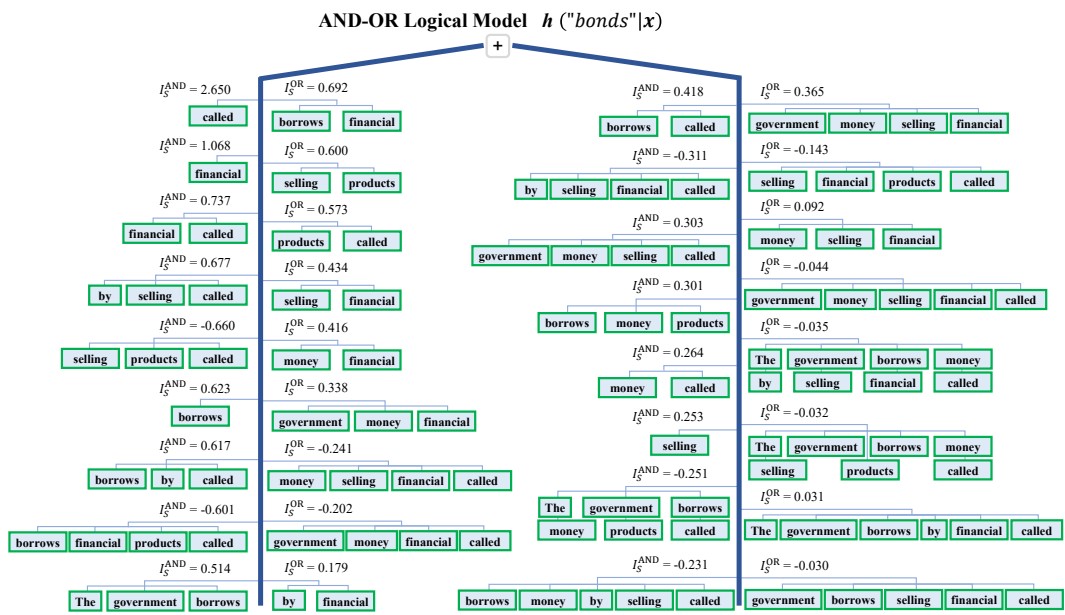

Figure 13: The logical model representing the inference logic of the original DeepSeek-R1-Distill-LLaMA-8B model on the input prompt "The government borrows money by selling financial products called." The predicted next word is "bonds."

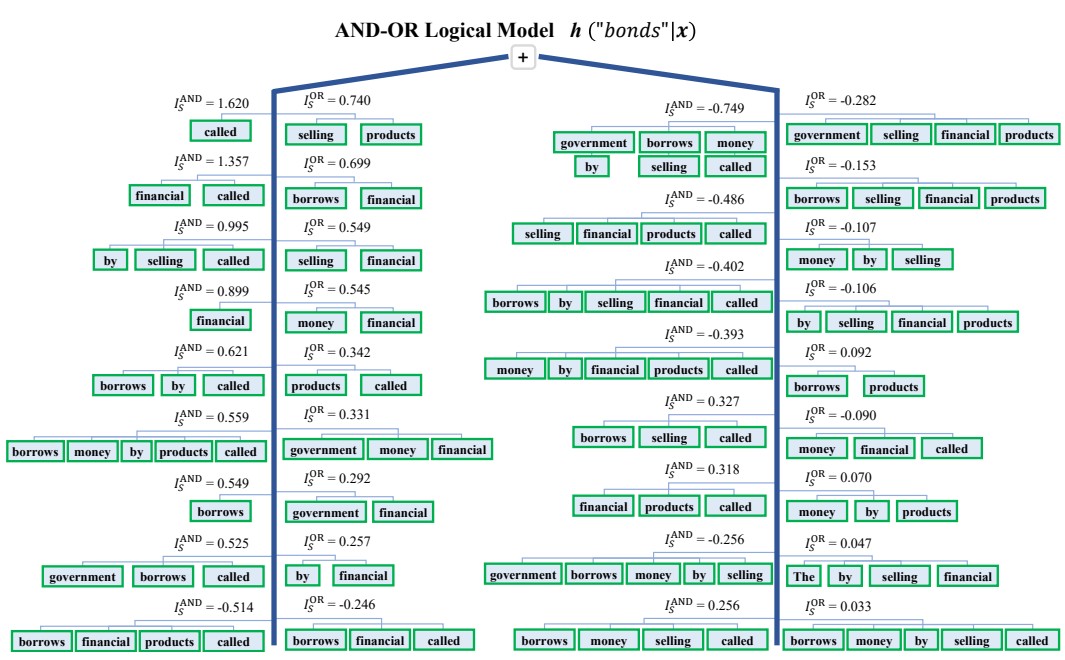

Figure 14: The logical model representing the inference logic of the pruned DeepSeek-R1-Distill-LLaMA-8B model (under a pruning ratio of 41.01) on the input prompt "The government borrows money by selling financial products called." The predicted next word is "bonds."

## J    DETAILED RESULTS OF THE THREE-PHASE DYNAMICS OF INTERACTIONS

### J.1    DETAILED RESULTS ON VGG-19

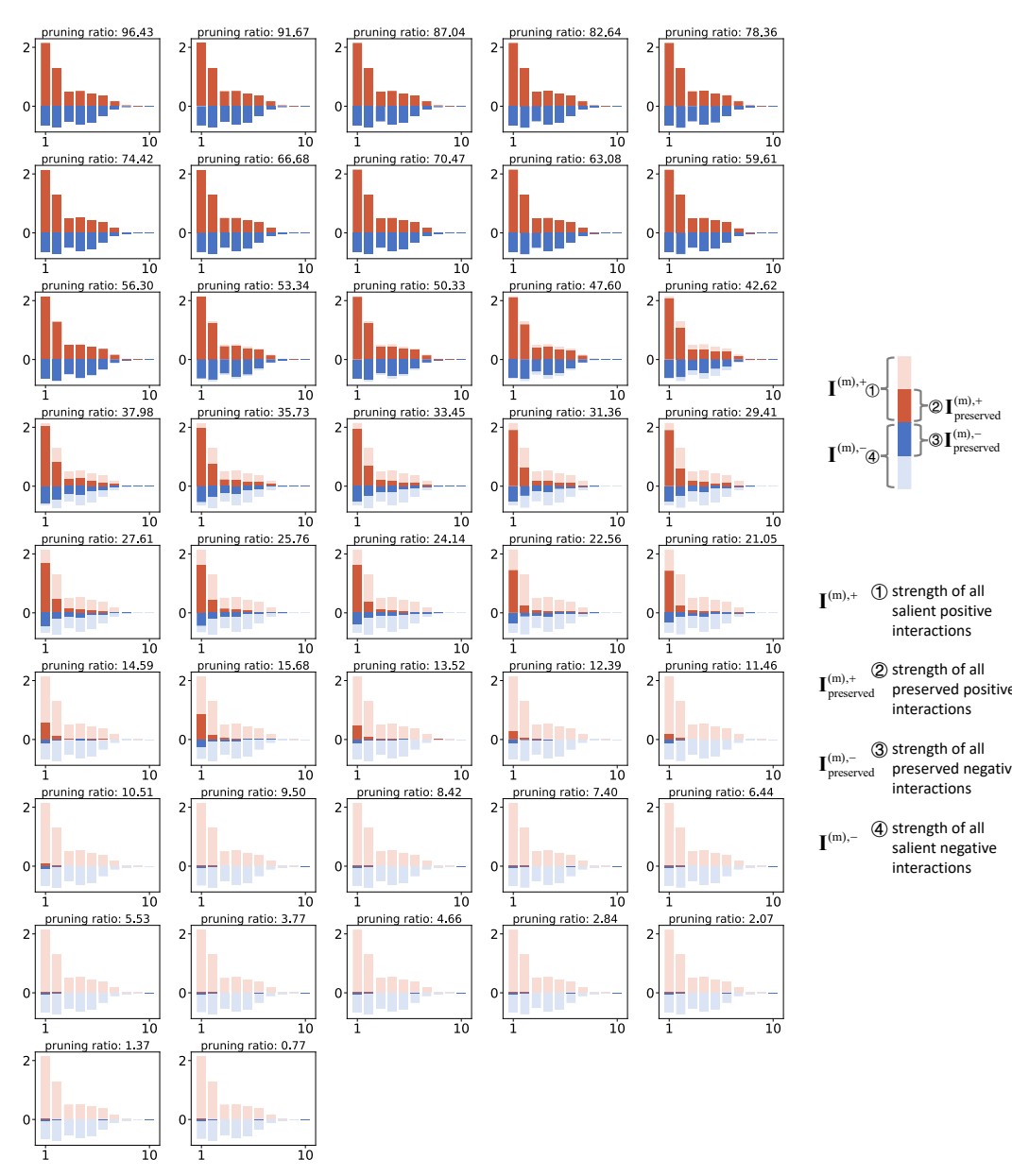

Figure 15: The distributions of interactions and preserved interactions across different orders encoded by pruned VGG-19 at different pruning ratios.

## J.2 DETAILED RESULTS ON RESNET-56

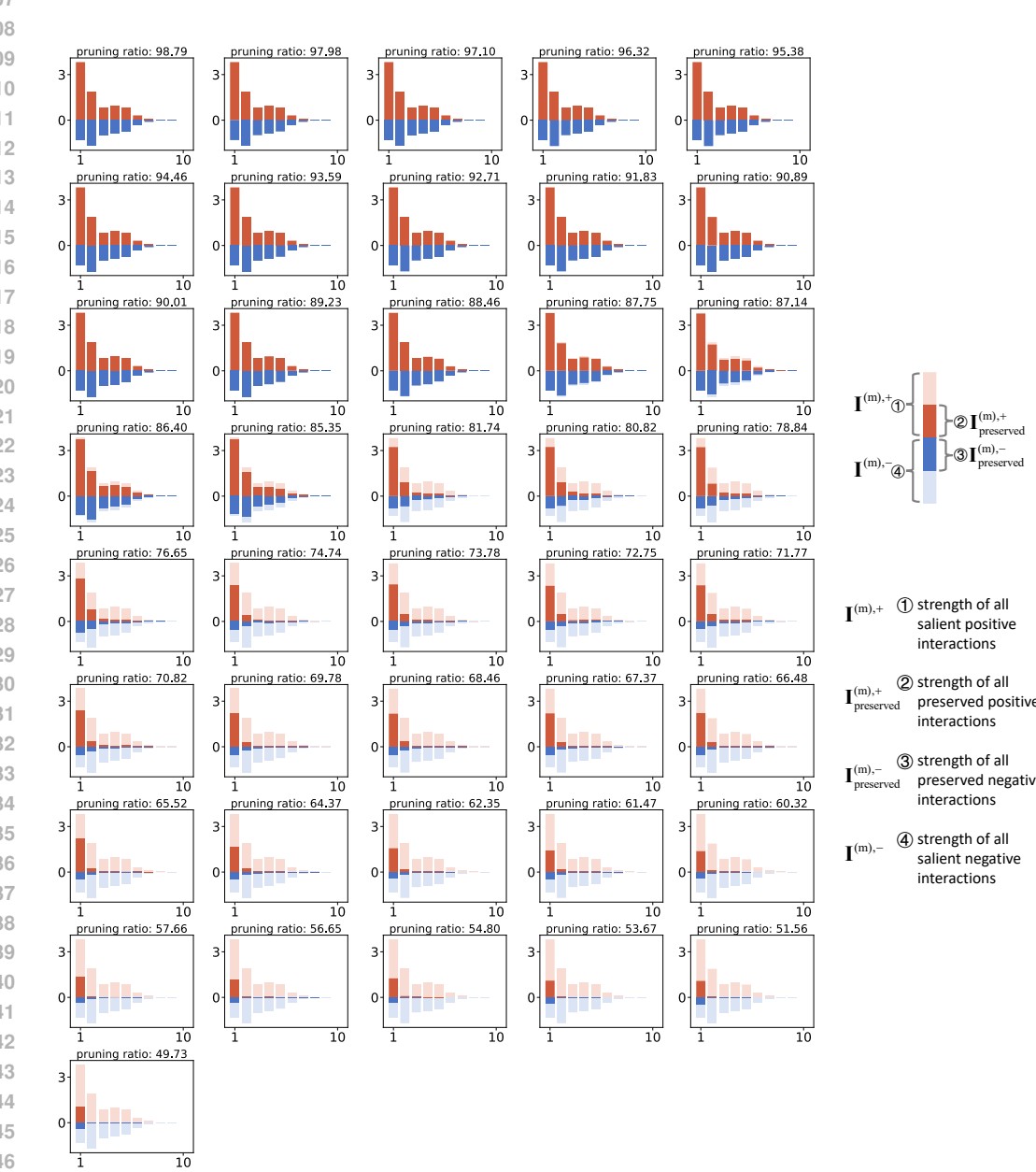

Figure 16: The distributions of interactions and preserved interactions across different orders encoded by pruned ResNet-56 at different pruning ratios.

## J.3 Detailed results on ResNet-50

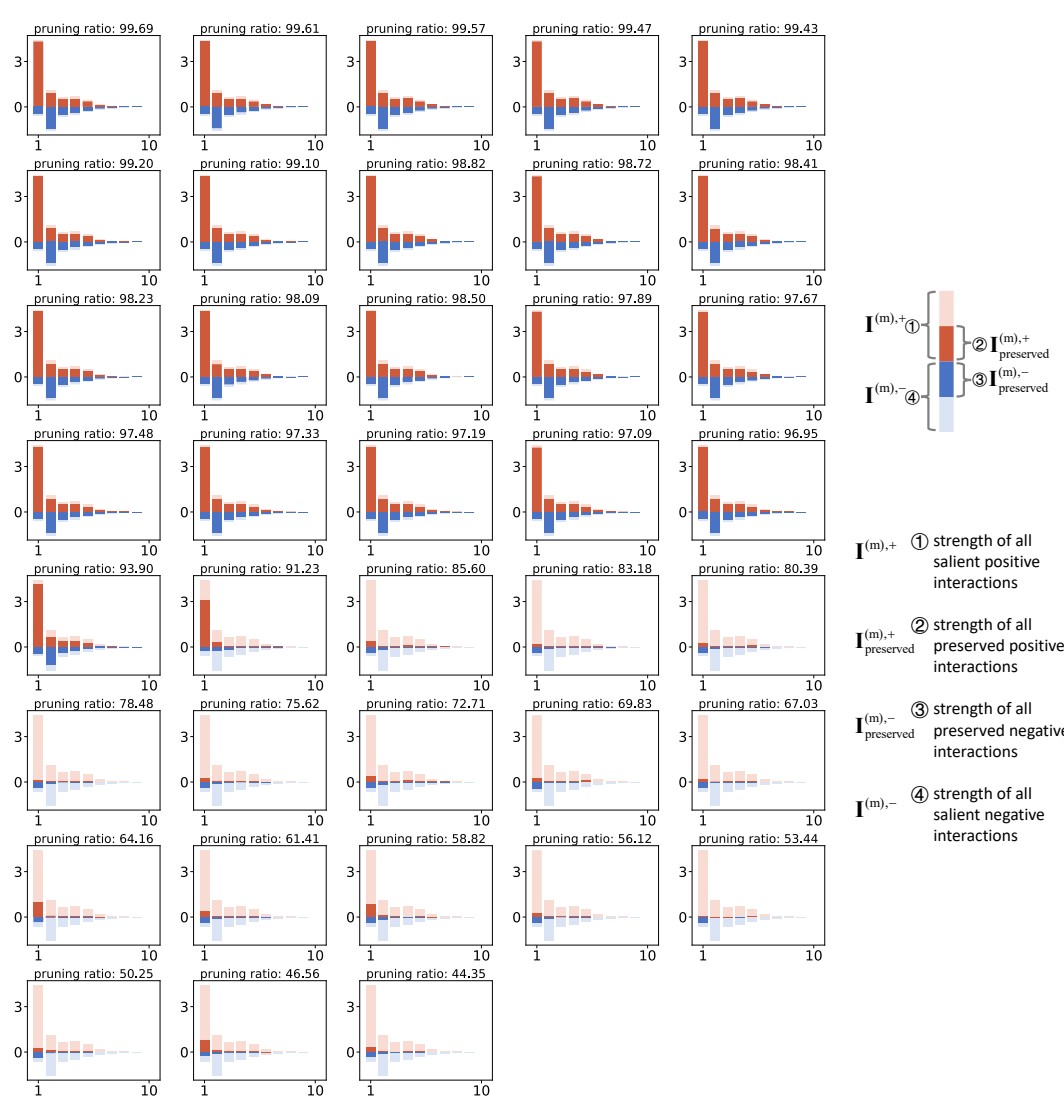

Figure 17: The distributions of interactions and preserved interactions across different orders encoded by pruned ResNet-50 at different pruning ratios.

## J.4 DETAILED RESULTS ON RESNET-101

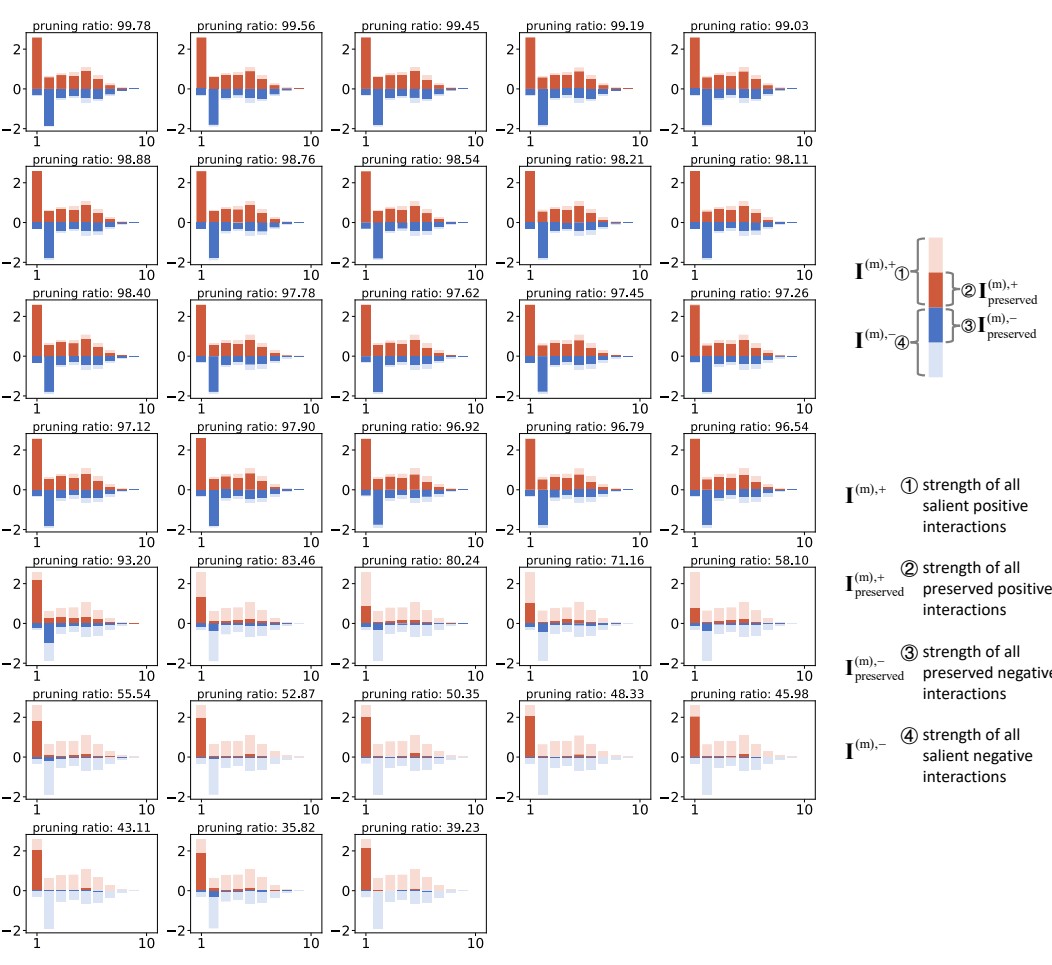

Figure 18: The distributions of interactions and preserved interactions across different orders encoded by pruned ResNet-101 at different pruning ratios.

## J.5 DETAILED RESULTS ON VIT-SMALL

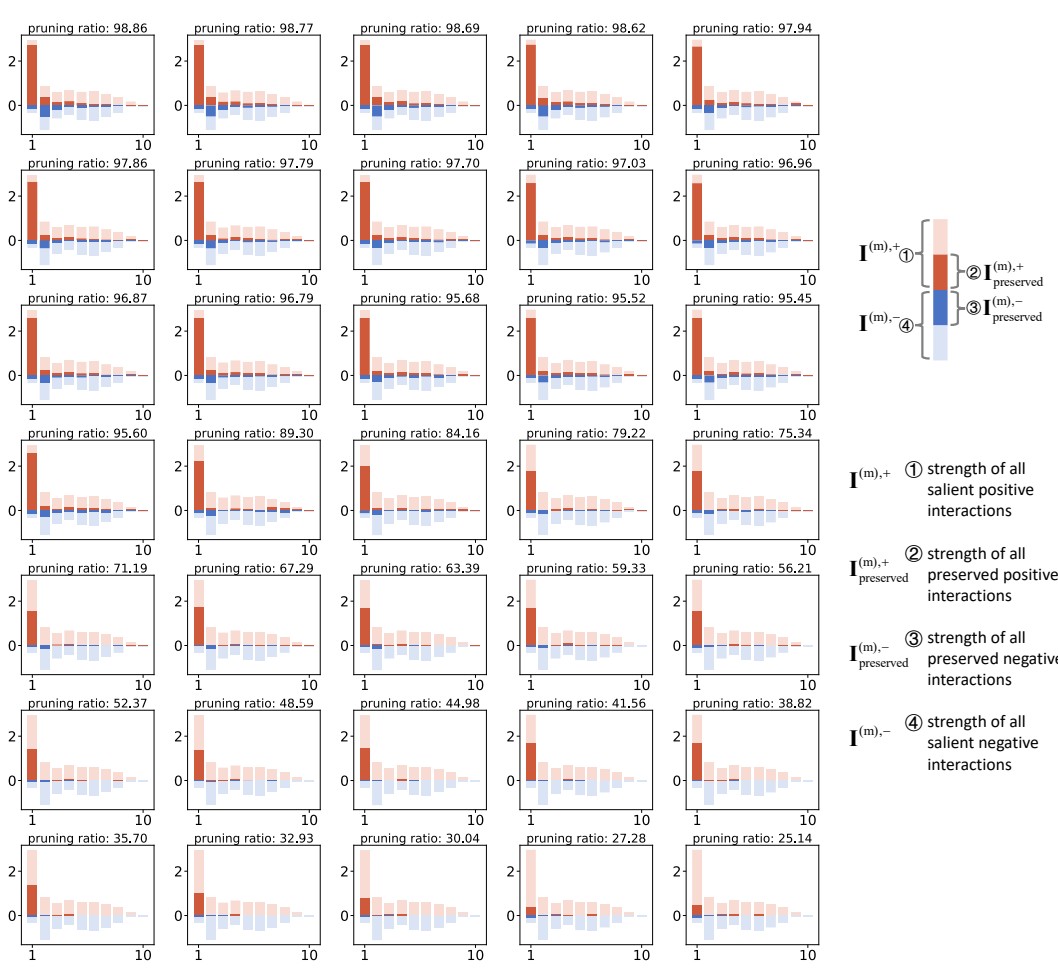

Figure 19: The distributions of interactions and preserved interactions across different orders encoded by pruned ViT-Small at different pruning ratios.

## J.6 DETAILED RESULTS ON VIT-BASE

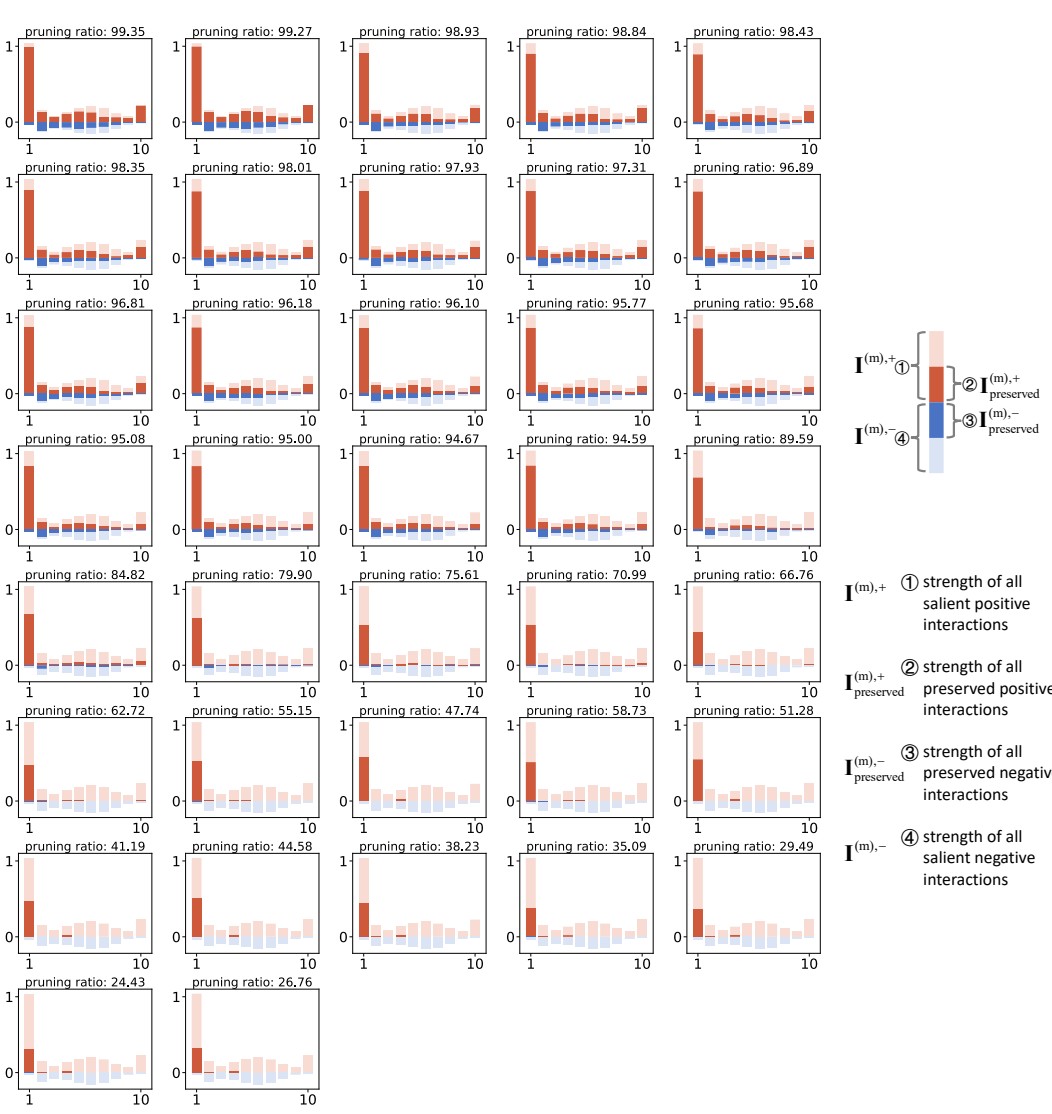

Figure 20: The distributions of interactions and preserved interactions across different orders encoded by pruned ViT-Base at different pruning ratios.

## J.7 DETAILED RESULTS ON DEEPSEEK-R1-DISTILL-LLAMA-8B

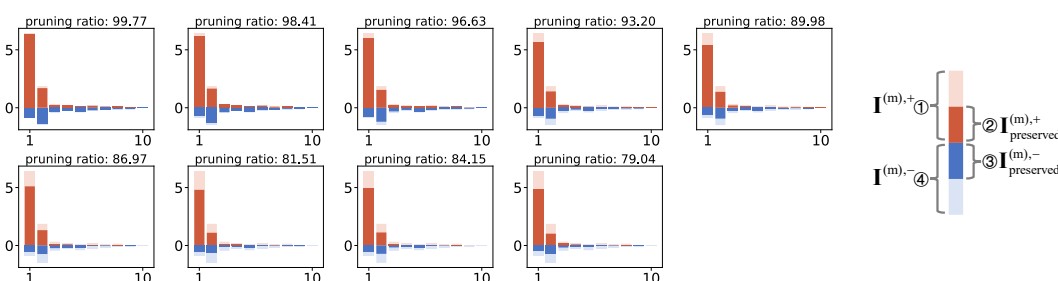

Figure 21: The distributions of interactions and preserved interactions across different orders encoded by pruned DeepSeek-R1-Distill-LLaMA-8B at different pruning ratios.

## J.8 DETAILED RESULTS ON LLAMA-3.1-8B

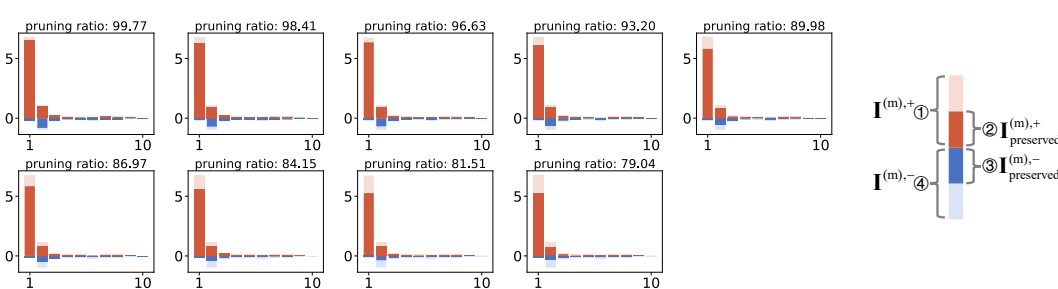

Figure 22: The distributions of interactions and preserved interactions across different orders encoded by pruned LLaMA-3.1-8B at different pruning ratios.

## J.9 RESULTS ON ALL EIGHT DNNS

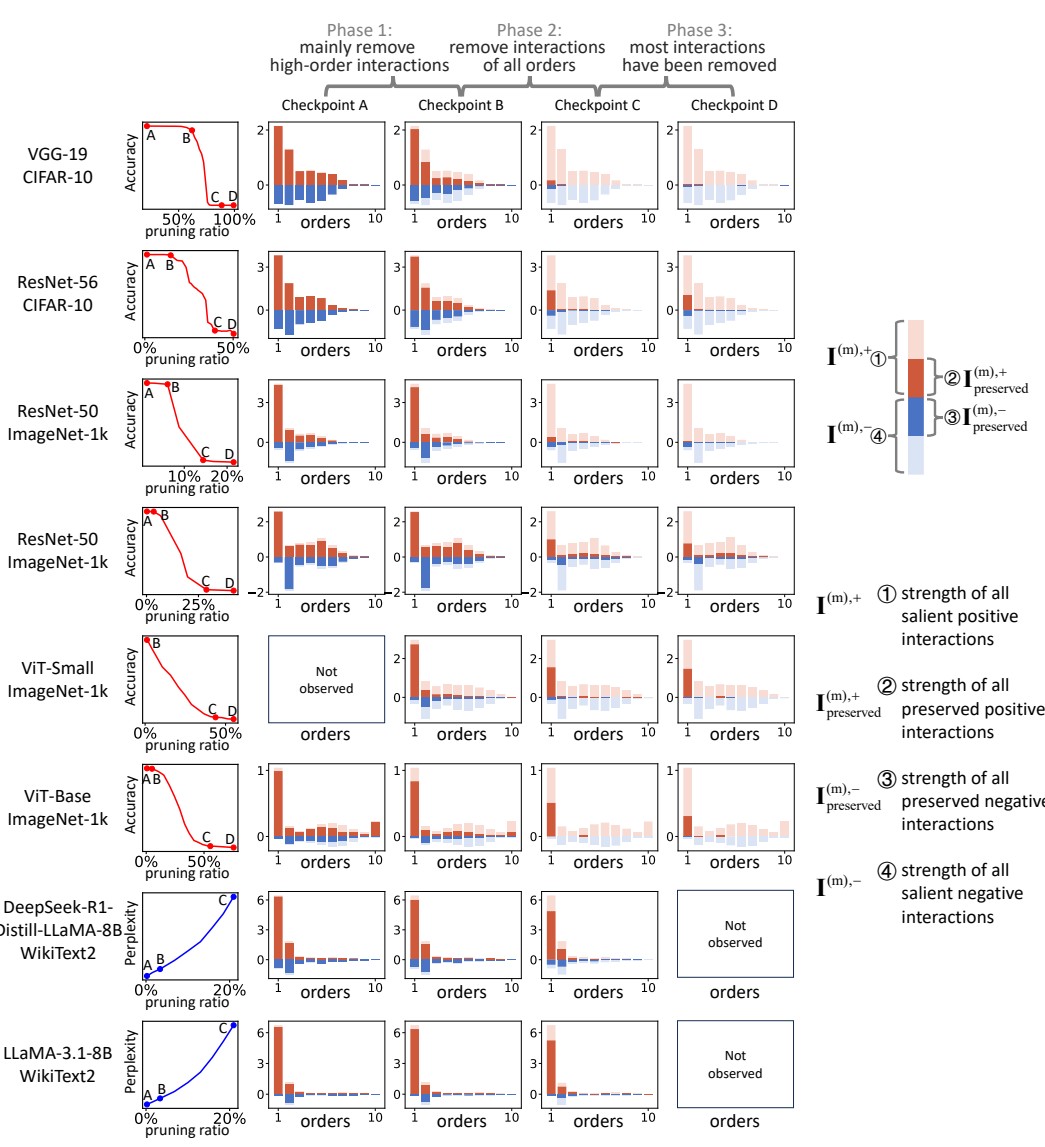

Figure 23: Changes of the distributions of $\mathbf{I}^{(m),+}$, $\mathbf{I}^{(m),-}$, $\mathbf{I}^{(m),+}_{\text{preserved}}$, and $\mathbf{I}^{(m),-}_{\text{preserved}}$ on eight DNNs when the pruning ratio increases. These changes can be divided into three phases.

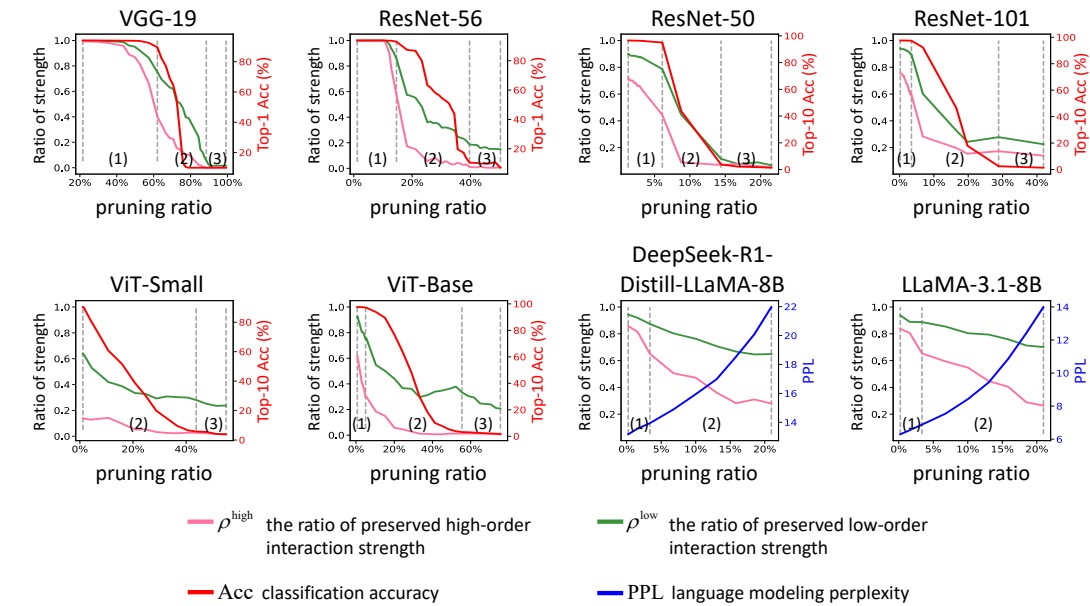

Figure 24: Changes of the ratio of preserved low-order interaction strength $\rho^{\text{low}}$, the ratio of preserved high-order interaction strength $\rho^{\text{high}}$ and classification accuracy (or language modeling perplexity) on eight DNNs when the pruning ratio increases.

Table 3: Efficiency of parameter pruning in the three-phases.

| Architecture | Phase 1 | Phase 2 | Phase 3 |
|---|---|---|---|
| ResNet-56 | $e^{1.21\to14.65} = 0.92$ | $e^{14.65\to39.68} = 0.78$ | $e^{39.68\to50.27} = 0.38$ |
| VGG-19 | $e^{21.64\to62.02} = 0.89$ | $e^{62.02\to88.54} = 0.61$ | $e^{88.54\to99.23} = 0.36$ |
| ResNet-50 | $e^{1.28\to6.10} = 0.91$ | $e^{6.10\to14.40} = 0.61$ | $e^{14.40\to21.52} = 0.72$ |
| ResNet-101 | $e^{0.22\to3.46} = 0.93$ | $e^{3.46\to28.84} = 0.75$ | $e^{28.84\to41.90} = 0.70$ |
| ViT-Small | — | $e^{1.14\to43.79} = 0.57$ | $e^{43.79\to55.02} = 0.27$ |
| ViT-Base | $e^{0.65\to5.00} = 0.87$ | $e^{5.00\to55.42} = 0.69$ | $e^{55.42\to75.57} = 0.55$ |
| DeepSeek-R1-Distill-LLaMA-8B | $e^{0.23\to3.37} = 0.78$ | $e^{3.37\to20.96} = 0.75$ | — |
| LLaMA-3.1-8B | $e^{0.23\to3.37} = 0.74$ | $e^{3.37\to20.96} = 0.70$ | — |

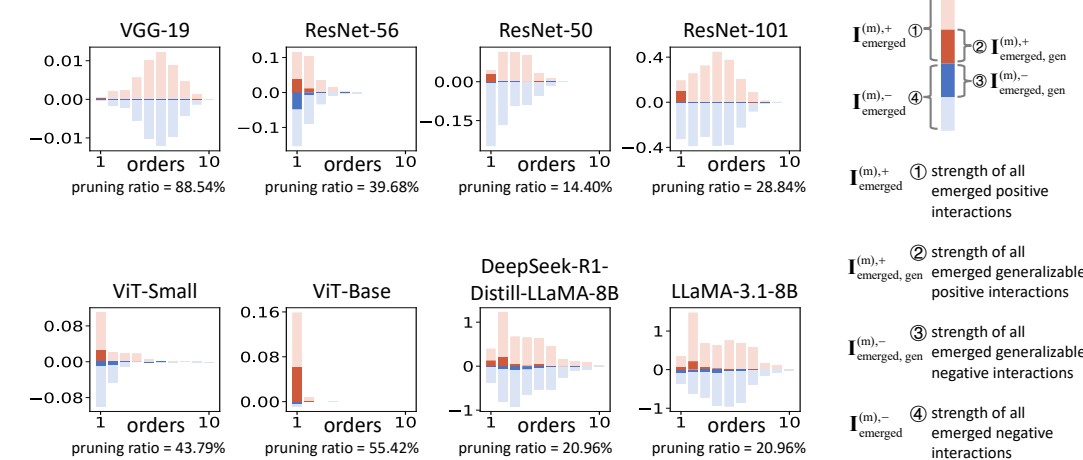

Figure 25: The distributions of the emerged interations introduced by the parameter pruning operation $\mathbf{I}^{(m),+}_{emerged}$, $\mathbf{I}^{(m),-}_{emerged}$, $\mathbf{I}^{(m),+}_{emerged,gen}$ and $\mathbf{I}^{(m),-}_{emerged,gen}$ on eight pruned DNNs. Emerged interactions are barely generalizable.

## K DISTRIBUTIONS OF GENERALIZABLE INTERACTION

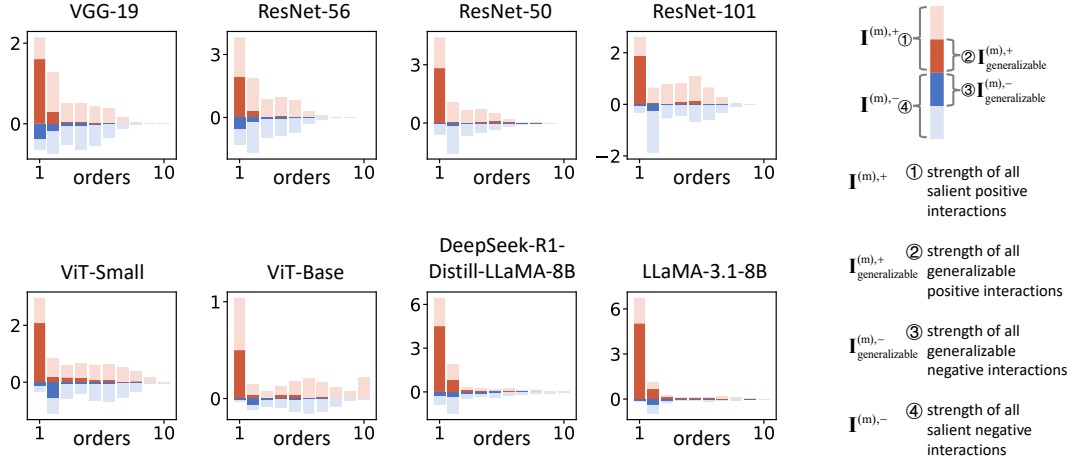

Figure 26: The distributions of interactions and generalizable interactions $\mathbf{I}^{(m),+}$, $\mathbf{I}^{(m),-}$, $\mathbf{I}^{(m),+}_{generalizable}$ and $\mathbf{I}^{(m),-}_{generalizable}$ encoded by eight original DNNs. Interactions of different orders exhibit different generalizability.

## L THE USE OF LARGE LANGUAGE MODELS (LLMs)

In this work, large language models (LLMs) were used solely as a general-purpose writing assistant to polish the grammar and improve the clarity of the text. No part of the research ideation, experiment design, data analysis, or substantive content generation relied on LLMs. The authors take full responsibility for the content of the paper.

