# OpenReview forum: "Towards the Dynamics of Representation Changes during Parameter Pruning for Network Compression"
_ICLR.cc/2026/Conference — Submitted to ICLR 2026_

### Official Review · Reviewer_Y2jP · 2025-10-30

**Soundness:** 1
**Presentation:** 2
**Contribution:** 2
**Rating:** 4
**Confidence:** 4

**Summary:**

This paper explore the dynamics of parameter pruning in neural networks through the lens of symbolic generalisability, they argue that the ration strength of low order interactions governs the accuracy fluctuations observed during pruning and that most often when a network is pruned and retains accuracy that this is due to the removal of high order interactions that respond only to 'noisy' features compared to the low order interactions which focus on generalisability. The paper conducts a multi-domain study examining multiple architectures across the vision and language domains. In their analysis, they identify three phases of pruning, which they argue govern accuracy reductions witnessed as the pruning ratio increases. The first phase is the identification of high-order interaction removal with a limited accuracy decrease, the second phase is characterised by the gradual removal of low and high-order interactions, which harms accuracy and the final phase is the majority removal of low-order interactions, which severely harms accuracy.

**Strengths:**

1. This paper uses the interesting lens of symbolic generalisability and extends it using an interventional study to explore the dynamics of increased pruning ratios on accuracy reduction, which identifies an interesting pattern between the ratio of strength between high-order interactions and low-order interactions and accuracy.
2. There is a good coverage of multiple modalities (vision and language) with a good representation of architectures for each modality. This helps to strengthen the findings and shows a degree of generality to the identified three phases of pruning.
3. There is an attempt (in Appendix B) to connect the observation of the importance of low-order interactions to practical insights which (somewhat) increase the robustness of a network to high pruning ratios.

**Weaknesses:**

Overall, considering the efficacy of a pruning strategy through the lens of low and high-order interactions is interesting, but from the results presented (apart from those in Appendix B), it feels hard to gauge what the practical insights of this work are. I do not consider a paper with limited practical insights to be a reason for rejection, but I feel as if this work would benefit from providing an insight into which pruning strategies are beneficial based on the low and high-order perspectives presented in this paper. Without this element, it is difficult to contextualise these findings among other work, such as that which helps to identify optimal sparsity values.

1. **No mean or standard deviations presented**: In all the figures and the tables, there is a lack of reporting the mean and the standard error of the mean, as a result, it is difficult to know the reliability of these results. For example, it is unknown if there is a high variability in the trends of the three phases without such statistics. Especially for the vision models (VGG-19 and ResNet-56), which are trained on CIFAR10, it should be possible to train multiple models. Additionally, it could be possible to train a smaller language model to ensure that robust statistics can be provided in the main body of the paper.

2. **Figure 3**: Why is it the case that for the language models (DeepSeek-R1 and LlaMA), when the perplexity increases largely between points B and C, the low-order interactions appear to be preserved? It would be my understanding that if, as stated in the paper  (L187), 'higher orders were usually less generalizable', that we would not expect such an impact from their removal on perplexity while the lower order 'generalisable' interactions remain. I see these two figures as counter evidence to the importance of lower-order interactions, which can be observed for the VGG19 and ResNet56 models. Also, if the low-order interactions are generalisable, then why, when we observe a complete degradation in accuracy at checkpoint D for the ResNet56, are some high-order interactions preserved?

Furthermore, for Figure 3, it would be useful if there were values on the y-axis and/or if the figures displaying the phases and the interaction distributions were larger.

3. **Comparison of Different Pruning Strategies**: Why do you not compare how efficacious different pruning strategies are to a specific architecture based on the low-order and high-order dynamics observed? It feels that this could add depth to the experiments and show a more practical use case for these insights.

4. **Ratio of Interaction Preservation (Figure 4)/Overstatements**: For the ResNet-56, when it hits 0% accuracy, a circa 0.2 ratio strength of low-order interactions remains. If low-order interactions are supposed to be 'generalisable', how can this be the case when the network can no longer generalise, and they remain? This gives me the impression of a correlation rather than a causal nature, as you suggest. Again, in this figure for both language models, even when the ratio of low-order interactions dominates at a pruning ratio of 20% there is a huge increase in zero-shot perplexity when the only large reduction is in the ratio of high-order interactions. How do you explain this?

5. **Identifying Optimal Compression Ratio**: It is unclear to me how exactly the optimal compression ratio can be identified more efficiently using the level of interactions. Could you provide an experiment when you accurately predict the optimal compression ratio (i.e. limited reduction in accuracy by the highest pruning amount) using this method/perspective? Also, how do you imagine that this is more efficient than simply evaluating accuracy with a forward pass of the test set if you need to train a reference model in the first place?

6. **Choice of VGG-16 and AlexNet (Appendix B)** Why did you choose the VGG-16 and AlexNet architectures to demonstrate the practicality of your findings to increase robustness to pruning? You do not use these architectures elsewhere in the paper? Could you please show the efficacy of the low-order optimisation method on the VGG-19 and ResNet-56, as well as on a language modelling task, to be consistent with the analysis conducted in the main paper?

**Questions:**

**Q1**: Given Figure 3's potential counter evidence to the importance of low-order interactions, is checkpoint D not recorded for Figure 3? It could provide good insight into whether there is a discrepancy between low and high-order interactions for these models.

**Q2**: Please can you further clarify how exactly you (L665) 'penalize the strength of all non-generalizable interactions during DNN training.'? Is this through a particular term in the loss, or via another mechanism?

**Q3**: Can you explain why there is a difference in Appendix B between the AlexNet and the VGG when using the high-order penalisation, as it appears that it is far more effective on the VGG over AlexNet.

**Q4**: How does your work relate to existing literature [6] on pruning that considers functional similarity as a core component of accuracy preservation over high and low order preservations?

**Q5** I am interested in the deviations of low-order and high-order interactions based on your base model, what is the consistency of interactions if you change architectures? For example for the CIFAR10 case you used the VGG19 and the ResNet-56 as each others reference model, how would you expect the results to change if you used the ViT as the reference model?


References:

[1] Petzka, H., Kamp, M., Adilova, L., Sminchisescu, C. and Boley, M., 2021. Relative flatness and generalization. Advances in neural information processing systems, 34, pp.18420-18432.

[2 Liang, T., Poggio, T., Rakhlin, A. and Stokes, J., 2019, April. Fisher-rao metric, geometry, and complexity of neural networks. In The 22nd international conference on artificial intelligence and statistics (pp. 888-896). PMLR.]

[3] Belia, S., Fidler, F., Williams, J. and Cumming, G., 2005. Researchers misunderstand confidence intervals and standard error bars. Psychological methods, 10(4), p.389.

[4] Wen, K., Li, Z. and Ma, T., 2023. Sharpness minimization algorithms do not only minimize sharpness to achieve better generalization. Advances in Neural Information Processing Systems, 36, pp.1024-1035.

[5] Mason-Williams, I., Ekholm, F. and Huszár, F., 2024. Explicit regularisation, sharpness and calibration. In NeurIPS 2024 Workshop on Scientific Methods for Understanding Deep Learning.

[6] Mason-Williams, G. and Dahlqvist, F., 2024, May. What makes a good prune? maximal unstructured pruning for maximal cosine similarity. In The Twelfth International Conference on Learning Representations.

---

### Official Review · Reviewer_npFo · 2025-10-30

**Soundness:** 1
**Presentation:** 1
**Contribution:** 2
**Rating:** 2
**Confidence:** 3

**Summary:**

This paper explores how pruning ratios affect the internal interactions within the model. Within this they use the perspective of symbolic generalisability to discover 3 phases within pruning, (1) `High` order interactions are removed (performance is maintained), (2) `High` and `Low` order interactions are removed (Performance degrades), (3) Both `High` and `Low` interactions hardly change as they are severely reduced (Performance is near or at Random). They suggest that because `High` order interactions are unlikely to generalize, they can be removed without harming the performance of the model. Whereas `Low` order interactions generalize, cannot be removed without harming performance.

**Strengths:**

Explores an interesting question and uses symbolic generalization in a unique way to explore this.

**Weaknesses:**

## Magic Numbers

1. Table 2: is filled with magic numbers with no explanation given. For example ResNet-56 phase 1 is $e^{1.21 \rightarrow 14.65}$ and VGG-19 is $e^{21.64 \rightarrow 62.02}$. Why are both not compared to the baseline model with zero pruning? That creates a fair comparison; otherwise, two values could be selected to explicitly show this value that is close to 1. Please explain the section of all the numbers in this table.

2. Figure 5: Why are these pruning ratios used: 88.54%, 39.68%, 20.96% and 20.96%?

## Missing Data

There is a lot of missing data with no rationale as to why:

1. Figure 3. The DeepSeek and LLaMA models' results for checkpoint D are not reported.

2. Figure 3. The models are all pruned to different amounts with no justification.

3. Figure 4: Phase 3 is not shown for DeepSeek and LLaMA. Why is this that?

4. Figure 4: The models are all pruned to different amounts with no justification.

5. Table 2: The efficacy parameter for phase 3 for the DeepSeek and LLaMA is missing. Why?

6. Table 3: The efficacy parameter for `Phase 1` for `ViT-Small` is not provided. Nore  is DeepSeek and LLaMA for `Phase 3`.

## Lack of Justification and Explanation

In Figure 4, both the VGG-19 and ResNet-56 have a ratio of 1.0 for preserved High and Low interactions when the pruning ratio is at 0%; however, for the DeepSeek and LLaMA models, these values do not start at 1 when the model has not been pruned. How is this the case? Surely all interactions are preserved before pruning, as that is the base model?

In Table 2 Efficiency of parameter pruning in the three phases. The ResNet-56 shows `Phase 1` at 0.92 and `Phase 2` at 0.78. VGG-19 shows `Phase 1` at 0.89 and `Phase 2` at 0.61. However, the DeepSeek and LLaMA models show `Phase 1` at 0.78 and 0.74, respectively. Would the efficiency parameter not suggest that the DeepSeek and LLaMA models are instead in `Phase 2`? Why have they been categorized as `Phase 1`?

## MISC

In Figure 15, the `Pruning Ratio is at 96.43%` and the model is essentially completely preserved. Is this meant to be $100-96.43= 3.56$%? This is also similarly the case for Figure 16-21. Given that in all the other figures, 0% pruning ratio results in the model with the best performance.

In Appendix B, the results are shown for Top-5 Accuracy. Why is this the case? Why not Top-1? Also, how are the low-order and high-order interactions penalized?

**Questions:**

See weakness, but more concretely:

Why are different compression ratios used in table 2 for different architectures?

What happens to the Efficiency Parameter over pruning? Could this be shown in 1% intervals from 0-100% and compared to the baseline for all models explored in the paper, such that $e^{0\rightarrow p}$  where p is the pruning ratio.

In Figure 4, both the VGG-19 and ResNet-56 have a ratio of 1.0 for preserved High and Low interactions when the pruning ratio is at 0%; however, for the DeepSeek and LLaMA models, these values do not start at 1 when the model has not been pruned. How is this the case? Surely all interactions are preserved before pruning, as that is the base model?

In lines `374-375`, the paper states, `Most generalizable interactions are low order interactions, and most high-order interactions cannot generalize to the reference DNN`. What is the ratio of low-order interactions that can and cannot generalize, and what is the case for high-order interactions?

---

### Official Review · Reviewer_Fy4c · 2025-11-01

**Soundness:** 3
**Presentation:** 2
**Contribution:** 2
**Rating:** 4
**Confidence:** 4

**Summary:**

This paper understands the parameter removal process from a model interpretability aspect, especially of the information intersection perspective. Specifically, it states the pruning operation will firstly remove parameter with high-order interaction since these parts are hard to be generalized and their final effect of model performance is neglectable. Along with the increasing pruning ratio, the other relatively important model parts are affected. Overall, it is a new perspective to treat model compression or pruning.

**Strengths:**

1. Using the and-or based interaction of model interpretability as an angle to understand pruning operation is relatively new to me.
2. This draft provides detailed theoretical analysis and ablation analysis.
3. There is also a proposed method based on the understanding above for model pruning practice.

**Weaknesses:**

1. If I understand correctly, the main goal of this paper is to state the pruning process can be seen from the interaction based model interpretability perspective. If so, I may concern the goal of this work is not concrete enough, which is more like a research finding or technical report. The conclusion itself is straightforward, high-order parameter (not necessary) will be removed first and more important one will be pruned later while the increasing pruning ratio.
2. Paper format needs improvement such as figures and ablation visualization. Too many bold font in the main text affects the readability.
3. The format in appendix is also not consistent. The draft needs polish to be ready with good readability.

**Questions:**

Please check above section.

---

### Official Review · Reviewer_NQqG · 2025-11-04

**Soundness:** 1
**Presentation:** 1
**Contribution:** 2
**Rating:** 2
**Confidence:** 4

**Summary:**

In this summary, I’ll try to clarify the mathematics underlying the paper as it is rather muddled in the submission. Hopefully this can clarify the discussions.

This paper is part of a family of papers looking at a particular class of decompositions of a function $v: \mathbb{R}^N\to \mathbb{R}$ implemented by a dnn (although this isn't necessary, any function will do). This decomposition is defined via the Möbius inversion formula applied to the finite poset $(\mathcal{P}(N), \subseteq)$, providing a generalised inclusion-exclusion perspective on $v$. Concretely, for a fixed function $v: \mathbb{R}^N\to \mathbb{R}$, input $x\in\mathbb{R}^N$, and masking function $m: \mathcal{P}(N)\times \mathbb{R}^N\to \mathbb{R}^N$, the Mobius inversion formula is applied to the map $\mathcal{P}(N)\to \mathbb{R}, T\mapsto v(m(T,x))$ which the authors denote as $v(x_T)$ and gives
$$ v(x_T) = \sum_{S\subseteq T} I^{and}(S,x)$$
iff
$$I^{and}(T,x)=\sum_{S\subseteq T} v(x_S) (-1)^{|T|-|S|}$$
If I understand the quite imprecise terminology, this would be called a decomposition in terms of "AND interactions".
In particular $v(x)=\sum_{S\subseteq N} I^{and}(S,x)$ (if we assume that $x_N=x$) and we can express $v$ as this sum of "AND interactions".

One can apply exactly the same theorem to the poset $(\mathcal{P}(N), \supseteq)$ and get.
$$ v(x_T) = \sum_{S\supseteq T} I^{or}(S)$$
iff
$$
I^{or}(T)=\sum_{S\supseteq T} v(x_S) (-1)^{|T|-|S|}=-\sum_{S\supseteq T} v(x_S) (-1)^{|S|-|T|}=-\sum_{N\setminus T\supseteq N\setminus S} v(x_S) (-1)^{|N\setminus T|-|N\setminus S|} = -\sum_{S\subseteq T} v(x_{N\setminus S}) (-1)^{|T|-|S|}
$$
Again, if I understand correctly, this would be called a decomposition in terms of "OR interactions" and $v$ can be written as a sum of OR interactions in the same was as it can be decomposed as a sum of AND interactions as described above.

The paper then presents a flat combination of these two decompositions (i.e. no nested logical structure) expressing $v$ in terms both of AND and OR components. This does not directly correspond to the Möbius inversion formula as some parameters must be learned/minimized (it may be worth looking at the poset $(\mathcal{P}(N)\times \mathcal{P}(N), \subseteq\times \supseteq)$ for a canonical combination).

Starting from this premise, the authors consider what happens to AND and OR interactions of various orders (where the order is the cardinality of $S$ in $I(S,x)$) under pruning. The authors claim that high-order interactions are removed first by pruning, meaning that they encode brittle, non-generalisable patterns. The definition of generalisable is given in terms of the AND/OR decomposition of a separate DNN trained on testing samples.

**Strengths:**

- The underlying idea behind the AND/OR decompositions behind this paper (and the others) is intuitively appealing (although it isn’t clear from Appendix I that it is useful or easy to interpret).
- The idea of using pruning to study this approach is also intuitively appealing.
- The working definition of generalisable works.
- The experiments are non-trivial and quite detailed (e.g. the study of emerging interactions conveys the impression that many possible aspects of the problem were looked at in some detail)
- Overall, there is potential and there is meat in this paper, and I would encourage the authors to improve it and keep at it

**Weaknesses:**

- This paper is part of a series of papers based on the same theorem (presented in different ways, it is Theorem 2 in this paper). I traced back the chain of references to Thm 1 of Ren, Jie, et al. "Defining and quantifying the emergence of sparse concepts in dnns." Proceedings of the IEEE/CVF conference on computer vision and pattern recognition. 2023 where the proof is given in an Appendix. It is a little worrying that so many papers can be published based on a result that may never have been peer-reviewed… However, as explained in the summary, it turns out that this result is a special case of a very well-known result called the Mobius inversion theorem. Theorem 2 should therefore refer to a standard textbook (e.g. Stanley, R. P. (2011). Enumerative combinatorics volume 1 second edition. Cambridge studies in advanced mathematics.) rather than this dubious chain of special cases.
- This being said, theorem 2 alters the statement of the result from previous work and introduces learnable parameters b, \gamma_T which are (a) not covered by any of the previous versions of the theorem and (b) not even covered by the proof in the Appendix G. In fact, the bias b does not really feature in the statement itself. Thus Theorem 2 is, as it stands, is highly suspicious. Moreover, either the parameters are learnable, in which case the theorem should contain some existential statement (there exist \gamma_T such that…), or they can be taken to be arbitrary, in which case they need not be learned.
- The fact that the proof of the OR part of theorem 2 (Appendix G) is not simply dual to that of AND is concerning as one should get one from the other by de Morgan (or, equivalently, by reversal of the poset order as shown above).
-The mathematics in this submission is generally well below par. Defining N={1,2,…,n} (line 120) or saying let A(r_1) be such and such and A(r_2) be such and such (l395) is forgivable for a 1st year undergrad, not for an ICLR submission. The notation is clunky (why use x’ when x would do? is \Psi in (2) the same as in line 157?) and many definitions are missing or imprecise (what is an “interaction”?, what does it mean for a subset of N to be present in an N-tuple? How can you “augment” an input by random masking, etc). The proof in Appendix G is a perfect example of how not to write a mathematical proof.
- The paper claims that it discovers “a specific three-phase dynamics of DNNs’ interaction changes under increasing pruning ratio”. Based on the experimental data presented in the paper, I’m not convinced by this claim. It feels a bit like saying that Goldilocks has discovered the three-phase dynamics of porridge temperature. Fig 3,4,6 essentially show that there are as many pruning ratio vs performance profiles as there are architecture-problem pairs.
- Appendix I is misleading: Fig 7-14 do not really provide a “logical structure”. The roots of the trees have no logical status, it's purely a display choice (this is quite misleading in fact), and the branches just list the AND and OR components in decreasing order of magnitude. This is not a “logical model”, in the same way as a decision tree would be, even though it misleadingly gives off that impression.

**Questions:**

Since AND decompositions and OR decompositions are perfect, in the sense that they make no error, why consider AND/OR decompositions? What is the advantage? And why have a flat AND/OR decomposition rather than a nested one which would echo conjunctive/disjunctive normal forms and provide some kind of “logical model”?

---

### Meta-Review · Area_Chair_NHHJ · 2026-01-05

**Summary:**

This paper studies pruning dynamics through an interaction-based interpretability lens and reports a three-phase pruning behavior. While the idea is interesting, reviewers consistently raise serious concerns about mathematical soundness, clarity, and experimental rigor. The theoretical formulation lacks a precise exposition and, in places, incorrect or unjustified. The main claims (especially the three-phase dynamics) lack proper evidence; and the experimental section contains choices that are not explained properly, missing experimental details, and inconsistent baselines.

Given the uniformly low scores, the lack of author rebuttal, and the unresolved issues, this submission does not meet the standards for ICLR in its current form.

**Reviewer Concerns:**

N/A (no rebuttal was submitted).

**Reviewer Scores:**

N/A (no discussion took place -- scores were uniformly low and there was no rebuttal from the authors).

---

### Decision · Program_Chairs · 2026-01-26

Reject